# EFFICIENT FINE-TUNING OF SINGLE-CELL FOUNDATION MODELS ENABLES ZERO-SHOT MOLECULAR PERTURBATION PREDICTION

## ABSTRACT

Predicting transcriptional responses to novel drugs provides a unique opportunity to accelerate biomedical research and advance drug discovery efforts. However, the inherent complexity and high dimensionality of cellular responses, combined with the extremely limited available experimental data, makes the task challenging. In this study, we leverage single-cell foundation models (FMs) pre-trained on tens of millions of single cells, encompassing multiple cell types, states, and disease annotations, to address molecular perturbation prediction. We introduce a drug-conditional adapter that allows efficient fine-tuning by training less than 1% of the original foundation model, thus enabling molecular conditioning while preserving the rich biological representation learned during pre-training. The proposed strategy allows not only the prediction of cellular responses to novel drugs, but also the zero-shot generalization to unseen cell lines. We establish a robust evaluation framework to assess model performance across different generalization tasks, demonstrating state-of-the-art results across all settings, with significant improvements in the few-shot and zero-shot generalization to new cell lines compared to existing baselines.

## 1 INTRODUCTION

Recent advancements in high-throughput single-cell RNA sequencing (scRNA-seq) have significantly deepened our understanding of cellular heterogeneity, enabling detailed insights into gene expression profiles and the dynamic responses of individual cells within complex tissues and microenvironments (Tanay & Regev, 2017; Han et al., 2020). Notably, the ability to study the biological impact of different perturbations, such as molecular treatments, provides a unique opportunity to unravel cell function and significantly accelerate biomedical research (Srivatsan et al., 2020). As a consequence, computational methods to predict cellular responses to perturbations hold promise as transformative tools for accelerating drug discovery and personalized medicine (Roohani et al., 2024; Rood et al., 2024; Bunne et al., 2024). Critical applications such as virtual screening, mechanism of action (MOA) identification, and target identification, all rely on the ability to characterize the complex, multifaceted effects of different drugs in the cellular environment.

Machine learning approaches for modeling outcomes of molecular perturbations have recently emerged in the literature, addressing the challenging task of predicting cellular responses $\tilde{x}$ (i.e., gene expression) given the initial cellular state $x$ (i.e., gene expression of control population, representing the underlying biological model system/cell line) and treatment $T$ (i.e., molecule). However, existing methods face several key limitations. First, several approaches are not able to generalize to novel perturbations (Lotfollahi et al., 2023), an essential capability for applications such as virtual screening. Ultimately, methods that can generalize to new molecules are hindered by extremely limited datasets, spanning hundreds of molecules across a few cell lines (Hetzel et al., 2022; Piran et al., 2024; Bereket & Karaletsos, 2024).

Indeed, predicting cellular responses to novel chemicals at single-cell resolution poses a *few-shot* challenge, as sample multiplexing techniques are both expensive and time-consuming, restricting the size and diversity of treatments and biological model systems (Srivatsan et al., 2020; McGinnis et al., 2019). Several methods have recently tackled this challenge. Proposed approaches include leveraging transfer learning from more available bulk RNA-seq data (Hetzel et al., 2022) or incorporating prior

knowledge of gene-gene associations into the model (Roohani et al., 2024). However, these methods are limited by the information explicitly encoded in the prior or available in the pre-training datasets. More importantly, these methods have focused on predicting responses to novel drugs or novel drug-cell-line pairs, instead of the more challenging task of *predicting the outcome of perturbations for unseen cell lines*.

To address these challenges, focusing on the few-shot prediction of molecular responses, including for unseen model systems (i.e., cell lines), we leverage emerging foundation models (FMs) (Bommasani et al., 2022). FMs have displayed notable success across several fields, including natural language, vision, and, more recently, biomedicine (Moor et al., 2023). In particular, single-cell FMs (Yang et al., 2022; Heimberg et al., 2024; Cui et al., 2024; Theodoris et al., 2023; Hao et al., 2024) are trained on tens of millions of single cells, encompassing multiple cell types, states, and disease annotations. Such pre-trained architectures have recently shown the potential to learn universal biological representations that can be adapted to specific tasks (Ma et al., 2024). By building on the zero-shot and few-shot capabilities of FMs (Wei et al., 2022), we aim to leverage the implicit knowledge of gene-gene relationships and cell states within single-cell FMs to characterize cellular responses to molecular perturbations.

Previous work explored the application of single-cell FMs for predicting outcomes of genetic perturbations (i.e., gene editing techniques) (Cui et al., 2024; Theodoris et al., 2023; Hao et al., 2024). However, such problem formulation is simplified by the fact that the treatment space (different genes) is the same as the response space, allowing direct fine-tuning of the pre-trained architecture. In contrast, modeling responses to chemical perturbations involves bridging cell representations with *a distinct modality* (i.e., molecular structures), making the task more complex. To address this challenge, we introduce a *drug-conditional adapter* layer which allows injecting molecular information into the trainable parameters, while leaving the original weights of the single-cell FM frozen.

In summary, single-cell Drug-Conditional Adapter (scDCA) efficiently fine-tunes a single-cell FM, linking it to a different modality (small molecules) to enable molecular property prediction. Our parameter-efficient approach allows fine-tuning on small chemical perturbation datasets, avoiding overfitting while preserving important background biological knowledge encoded in the pre-trained weights.

Through this strategy, the proposed method allows not only the prediction of cellular responses to novel drugs, but also the zero-shot generalization to unseen contexts (e.g., cell lines or cell types). Our contributions include:

- We investigate the fine-tuning of single-cell FM for molecular perturbation prediction, introducing scDCA, which utilizes efficient drug-conditional adapters.

- We extend the evaluation of this model beyond novel drug and drug-cell-line prediction (focus of prior work) to the more challenging task of unseen cell line prediction, which requires generalization to new biological contexts.

- We demonstrate state-of-the-art performance of scDCA, both in the prediction of unseen drugs and especially for unseen cell lines. By leveraging the rich biological representations learned from single-cell FMs and incorporating drug-conditional adapters, our approach enables robust predictions even in zero-shot settings, making it a powerful tool for drug discovery and cellular modeling.

## 2 RELATED WORK

**Prediction of perturbation experiments.** In this work, we address the biological challenge of predicting the transcriptional cellular responses to novel molecular perturbations. There have been a few studies that tackle this problem directly or indirectly.

Several methods focus on predicting the effects of genetic perturbations, where perturbagens correspond to genes and their combinations e.g., (Roohani et al., 2024). These approaches are not always directly applicable to the prediction of chemical perturbations, which include the vast chemical space (estimated to encompass up to $10^60$ drug-like compounds (Bohacek et al., 1996).

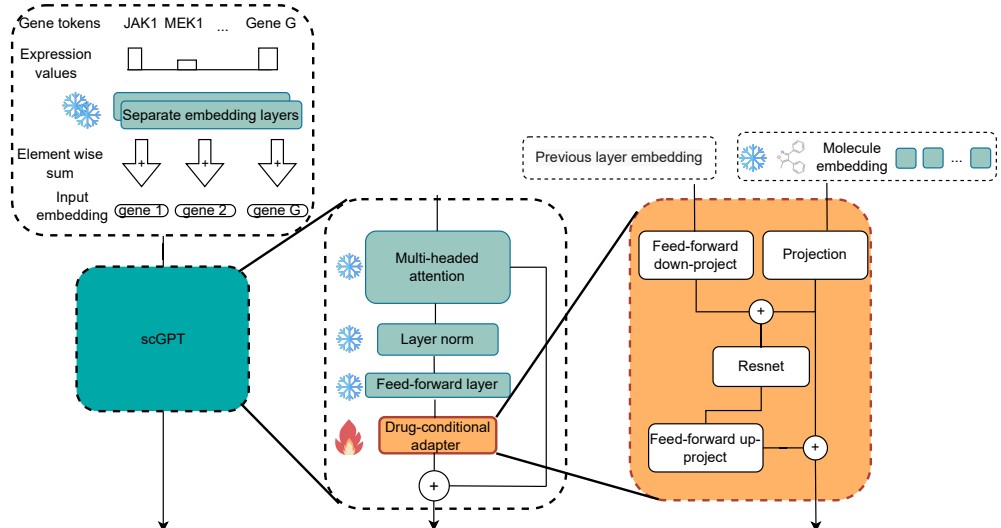

Figure 1: scDCA architecture. On the upper left, we show the input embedding to scGPT, which consists of gene tokens and unperturbed gene expressions. Input is passed through scGPT, which consists of stacked transformer blocks, where each layer incorporates a drug-conditional adapter module. Primary goal of this adapter is to introduce parameter-efficient fine-tuning by leveraging molecule embeddings to dynamically adjust biases of the down-projection and up-projection layers. Weights of the original transformer layers are frozen to reduce the number of trainable parameters.

ChemCPA (Hetzel et al., 2022) leverages an encoder-decoder architecture with separate embeddings for the perturbation (molecule) and the cell line, combined with an adversarial classifier that is used to generate invariant basal states and produce disentangled representations. Biolord (Piran et al., 2024) is a deep generative framework designed for learning disentangled representations in single-cell data. It partitions the latent space into subspaces and optimizes them to represent cell lines and perturbations, recombining them during inference. GEARS (Roohani et al., 2024) leverages prior knowledge of gene-gene interactions to predict outcomes of genetic perturbations. The Sparse Additive Mechanism Shift Variational Autoencoder (SAMS-VAE) (Bereket & Karaletsos, 2024) extends prior work (Lachapelle et al., 2022; Lopez et al., 2023) and employs an additive conditioning approach.

While SAMS-VAE focuses on genetic perturbation, it can also be extended to predict molecular perturbations.

**Single-cell foundation models.** Recently, there has been significant and active research on single-cell FMs (Yang et al., 2022; Heimberg et al., 2024; Cui et al., 2024; Theodoris et al., 2023; Hao et al., 2024). These models often employ the self-attention transformer architecture (Vaswani et al., 2017) to efficiently learn single-cell representations across extensive datasets, which can then be fine-tuned for numerous downstream tasks. scBERT (Yang et al., 2022) is a recent example within this category, closely mirrors BERT's (Devlin et al., 2019) pre-training strategy. Specifically, scBERT is pre-trained through an imputation task on 1.12 million human cells, where masked gene expression values are predicted based on the embeddings of all other genes within a cell. scGPT (Cui et al., 2024) leverages a generative transformer architecture pre-trained on a vast and diverse repository of over 33 million cells. This model effectively extracts critical biological insights relevant to genes and cells, as demonstrated by its downstream performance on multiple tasks, including cell type annotation, multi-omic integration, and gene regulatory network inference. scGPT has also been used to predict outcomes of genetic perturbations, where the perturbagens are different (combinations of) genes. However, given that the model has been retained solely on single-cell omics data, it is not directly applicable to drug discovery questions. Indeed, predicting outputs of chemical perturbations (such as drug-induced cellular responses, tissue-specific effects, and pharmacodynamics) requires modeling the effects of a different modality (i.e., chemical structures) not seen during training.

**Efficient fine-tuning of foundation models.** The ability of FMs to quickly adapt to new tasks (Bommasani et al., 2022) has sparked considerable interest in developing efficient fine-tuning techniques (Han et al., 2023). For example, prefix tuning (Li & Liang, 2021) refers to the technique of prepending a trainable tensor to each transformer block, allowing adapting to a new task without modifying the original model parameters. Li & Liang (2021) showed that prefix tuning achieves comparable results compared to fine-tuning all layers while using only 0.1% of the parameters of the original model. Adapters-based approaches (He et al., 2022; Lei et al., 2023) involve the insertion of small adapter layers within transformer blocks. Only the parameters of the adapters are trainable during fine-tuning. Adapters usually include down- and up-projections, which reduce the number of trainable parameters. In particular, our work is related to (Karimi Mahabadi et al., 2021), which introduces a multi-task adapter for NLP applications, where the parameters of the adapter depend on the task embedding. In our work, the adapter's parameters are instead conditioned on a different modality (chemical structures), unseen during pre-training. The concepts of prefix tuning and adapters have been further extended to recent large language models, such as LLaMA-Adapter (Zhang et al., 2024), a parameter-efficient fine-tuning method specifically designed for LLaMA. Other efficient fine-tuning techniques include pruning (Lawton et al., 2023) and reparametrization techniques (Hu et al., 2022). Efficient fine-tuning of foundation models is a rapidly evolving field encompassing diverse approaches. For a more detailed overview, we refer readers to recent surveys Han et al. (2023); Xin et al. (2024).

## 3 FINE-TUNING SINGLE-CELL FM WITH DRUG-CONDITIONAL ADAPTER

The proposed strategy aims to simultaneously address two challenges in fine-tuning single-cell FMs for molecular perturbation prediction: 1) the extremely limited amount of paired data linking molecules to cellular responses, and 2) the existence of a *different modality* (i.e., molecular structures) unseen during pre-training. While recent work has focused on the efficient fine-tuning of foundation models reducing the number of trainable parameters (Hu et al., 2022; Zhang et al., 2024), the fine-tuning of single-cell FMs has been limited to scenarios where inputs and/or outputs are cell states, without considering different modalities.

Although scDCA is generally applicable to transformer-based single-cell FMs (Yang et al., 2022; Cui et al., 2024; Theodoris et al., 2023; Hao et al., 2024), which constitute the vast majority of models in this category, in the following, we focus on *scGPT* (Cui et al., 2024) to describe the methodology and the experiments. To this end, in this section we also provide a concise overview of the scGPT framework.

### 3.1 PRELIMINARIES

**Problem definition.** We consider a dataset comprised of single-cell gene expression data following molecular perturbation, $\mathcal{D} = \{(X^i, d^i, c^i)\}_{i=1}^N$, where $i$ denotes the $i$th cell out of a total of $N$, $X^i \in \mathbb{R}^n$ describes its $n$-dimensional gene expression, $d^i \in \{\text{drugs in } \mathcal{D}\}$ describes the drug treatment it received, and $c^i \in C$ describes its cell type or cell line, with $C$ being the set of all cell lines. In practice, we mean-aggregate the profiles of cells with the same drug-cell-line combination, in effect treating them as "pseudobulk" measurements or "metacells" $X^{(d)}(c)$ for every drug-cell-line combination $(c, d)$. In particular, as input for for all models considered here, we featurize the cell lines through the control gene expression vector $X^{(0)}(c^i) \in \mathbb{R}^n$, calculated as the mean gene expression over all cells of cell line $c^i$ receiving the negative control perturbation in the dataset (usually dimethyl sulfoxide (DMSO)). In the following, we omit the superscript and denote a generic input profile by $(X, d, c)$.

In this work, to assess the generalization ability of different models, we explore various tasks where we hold out parts of all available drug-cell-line combinations and predict the *perturbed gene expression* using the available data (Figure 2). In the *unseen drug* task, all cell lines are seen during training, but a subset of drugs is reserved for testing (Figure 2a). This formulation corresponds to a traditional virtual screening problem. In the *unseen drug-cell-line* task, the training set excludes random combinations of cell lines and drugs during training, such that both individual molecules and cell lines can appear in different combinations in the training set (Figure 2b). This formulation corresponds to a matrix completion problem. In the *unseen cell line (few-shot)* task, cell lines are split into a training $C_{\text{tr}}$ and test $C_{\text{te}}$ set, and only a small fraction of drugs from the latter is observed

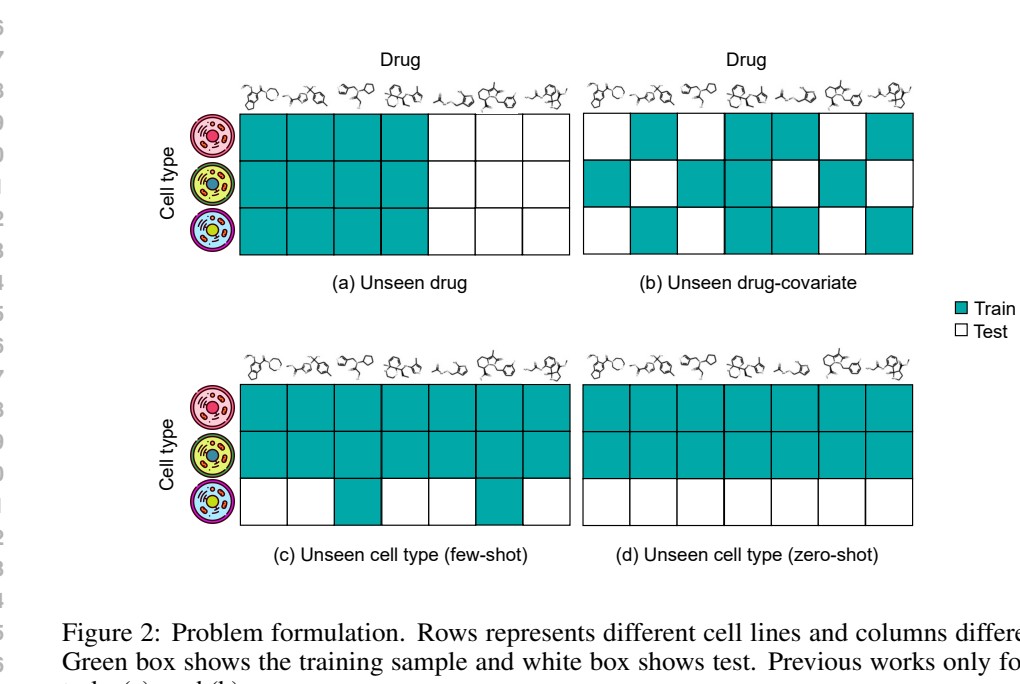

Figure 2: Problem formulation. Rows represents different cell lines and columns different drugs. Green box shows the training sample and white box shows test. Previous works only focused on tasks (a), and (b).

during training (Figure 2c). Finally, in the *unseen cell line (zero-shot)* task, we split cell lines as in the unseen cell-line (few-shot) task, but no observations from $C_{\text{te}}$ are available during training, and the model needs to be able to generalize to unseen cell lines based solely on the control gene expression (Figure 2d).

**Transformer-based single-cell FM architecture.** scGPT (Cui et al., 2024) relies on a transformer architecture, where the expression level of each input gene is treated as a token and a stack of self-attention transformers operates on all tokens corresponding to a single cell. Thus, in the scGPT framework, each gene is treated as the fundamental unit of information, similar to how a word functions in natural language generation.

More precisely, let us denote an individual cell profile by $(X, \text{cond})$ with expression level $X_g$ and gene-specific condition identifier $\text{cond}_g$ for every gene $g$. Here, the condition identifier $\text{cond}_g$ can incorporate a wide range of meta-information related to each gene. For example, for the task of genetic perturbation, the condition tokens would be the perturbed genes. In scGPT, first, the expression levels are discretized to integers $x_g$ by binning. Next, the identifiers for gene identity $t_g$, discretized gene expression $x_g$, and (discrete) conditions $\text{cond}_g$, are transformed into vectors in $\mathbb{R}^D$ by embedding layers $\text{emb}_g, \text{emb}_x, \text{emb}_{\text{cond}}$, respectively. The embeddings are then sum-pooled and concatenated into an initial cell tensor

$$h_0 = \text{emb}_g(t_g) + \text{emb}_x(x) + \text{emb}_{\text{cond}}(\text{cond}) \in \mathbb{R}^{M \times D}. \tag{1}$$

Finally, a stack of $L$ self-attention transformers operates on $h_0$, to obtain the final cell-specific gene embedding tensor $h_L$,

$$h_l = \text{transformer}_l(h_{l-1}) \quad \text{for } l = 1, \dots, L. \tag{2}$$

The model is pre-trained using single-cell RNA sequencing (scRNA-seq) data derived from public cell atlases. Similar to other single-cell omics FMs, scGPT uses a masked-attention transformer for generative training. Genes are randomly masked in the input sentence (akin to masking words in natural language) and reconstructed as output. The model is trained to minimize the mean squared error (MSE) loss,

$$\mathcal{L}_{\text{pre-train}} = \frac{1}{|\mathcal{U}_{\text{unk}}|} \sum_{g \in \mathcal{U}_{\text{unk}}} \left( \text{MLP}\left(h_L^{(g)}\right) - x_g \right)^2, \tag{3}$$

where $\mathcal{U}_{\text{unk}}$ denotes the set of masked genes, $h_L^{(g)}$ is the final scGPT gene token for gene $g$, MLP is a trainable multi-layer perceptron (MLP) layer, and the $x_g$ is the gene expression to be predicted.

**Molecular FM.** There have been several studies on building molecular FMs. In this work, we use ChemBERTa (Chithrananda et al., 2020), a molecular FM based on (Ro)BERT(a), pre-trained on 77 million compounds from PubChem (Kim et al., 2023). The input to this model is the SMILES structure of the molecule, and the output is an embedding vector. However, the proposed scDCA is compatible with virtually all molecular embedding models.

## 3.2 FINE-TUNING SCGPT

Compared to training a model from scratch for a new task, adapting and fine-tuning a FM for a new task using data from the target domain often leads to superior results (Wei et al., 2022; Wang et al., 2022; Shen et al., 2022; Wang et al., 2022). Various methods can be employed for this adaptation. In the following, we discuss several options, before introducing scDCA.

One commonly used technique is the *feature-based* approach, which leverages the embeddings of a model as features for training a separate downstream model. However, as demonstrated in Section A.4, this approach is suboptimal in our setting, primarily because the original single-cell FM was pre-trained exclusively on single-cell omics data across a wide range of conditions and cell types, while the target task involves molecular perturbations, resulting in subtle but critical shifts in gene expression that were likely not seen by the model at pre-training time.

Another approach involves *fine-tuning* the FM itself on the target dataset—in this case, readouts of molecular perturbations. To pursue this approach, we modify scGPT as follows. Following scGPTs approach to genetic perturbation prediction, we replace the discretization of input expression values with log1p-normalized expression values that are tokenized with a feed-forward network $\text{emb}_X$. In the inital embedding layer of scGPT, instead of the condition token, we incorporate a molecular embedding, broadcast across all genes as part of the input. Moreover, we provide the gene expression vector of the control perturbation on cell line $c$ as the gene expression input. Combined, denoting the molecular embedding by $\text{emb}_m(d)$ for a drug $d$, we change the initial embedding (1) to:

$$h_0 = \text{emb}_g(t_g) + \text{emb}_X(X^{(0)}(c)) + \text{emb}_m(d). \tag{4}$$

For the molecular embedding function $\text{emb}_m$, we employ ChemBERTa (Chithrananda et al., 2020), described in Section 3.1. The transformer stack of scGPT is then applied to $h_0$ as in (2) and processes it to output the perturbation outcome, employing a loss function similar to (3), but operating on log1p-normalized expression values $X$.

Although recent studies (Howard & Ruder, 2018) indicate that fine-tuning the entire model often yields superior performance, we find that this naive approach exhibits limited performance across various tasks (Section 4). We attribute this limitation to two key factors: 1) the extremely limited number of samples compared to the parameters of the pre-trained model, which can easily result in overfitting, and 2) the large domain shift resulting from adding molecule embeddings $\text{emb}_m(d)$ to the model's input, given that the pre-trained model has been solely trained to understand gene expression profiles.

To solve these challenges, we update the fine-tuning strategy by introducing our more parameter-efficient fine-tuning approach leveraging drug-conditional adapters, scDCA.

**Drug-Conditional Adapter.** Compared to the standard fine-tuning approach described above, in scDCA, the weights of the original network $\theta$ are frozen, and only the parameters of small adapter layers $\theta'$ are trained during fine-tuning. The goal of the adapter layers is twofold: minimize the number of trainable parameters, while giving the flexibility to generalize to different tasks particularly in a zero-shot setting.

Figure 1 summarizes our architecture. In the top left, we show the input to scGPT, which consists solely of the gene expression vector of the control perturbation in cell line $c$. Contrary to before, we do not pass the molecule embedding directly as input to scGPT to avoid overfitting. Instead, we directly pass the molecular embedding to adapter modules that we add to every transformer layer. This ensures that the input to the model during fine-tuning remains similar to the input used during pre-training.

The drug-conditional adapter modules (Figure 1 bottom right) consist of four main components: a molecular projection layer, a down-projection layer, a residual block, and an up-projection layer. Critically, we introduce a molecular projection layer that generates biases for the down-projection

and up-projection layers based on the molecule embeddings. Formally, let $h_l$ be the hidden state output from the $l$th transformer layer. To incorporate the influence of the molecular structure on the gene expression, we generate a bias vector $b_l = f_l^m(\text{emb}_m(d))$ through a module $f_l^m$ applied to the molecule embeddings. In our implementation, $f_l^m$ is modeled as a two-layer neural network. The adapter module processes $h_l$ as follows:

$$h_l^{\text{down}} = W_l^{\text{down}}h_l + \Pi^{\text{down}}b_l, \quad h_l^{\text{res\_net}} = \text{res\_net}_l(h_l^{\text{down}}), \quad h_{l+1} = h_l^{\text{up}} = W_l^{\text{up}}h_l^{\text{res\_net}} + b_l, \quad (5)$$

where $W_l^{\text{down}} \in \mathbb{R}^{d_{\text{bottleneck}} \times d_{\text{input}}}$ and $W_l^{\text{up}} \in \mathbb{R}^{d_{\text{input}} \times d_{\text{bottleneck}}}$ are the down-projection and up-projection weight matrices, respectively, with $d_{\text{bottleneck}}$ being the hidden state dimensionality and $d_{\text{input}}$ the input dimensionality, and $\Pi^{\text{down}}$ denotes projection to the first $d_{\text{bottleneck}}$ dimensions. During fine-tuning, trainable parameters are learned through a similar loss function as 3.

This design has critical advantages. First, it allows the model to dynamically adjust its parameters using molecule embeddings in a lower-dimensional space, significantly reducing the number of trainable parameters. Additionally, it avoids substantial input distribution shifts, as the model's input remains solely gene expression profiles (as seen during pre-training). Finally, it enables molecular conditioning, which is required for the task.

**Generalization to unseen drugs and cell lines.** By using a frozen molecular structure encoder, we leverage pre-trained embeddings with prior information about structural similarity. Self-supervised molecular representations have been previously shown to boost few-shot molecular property prediction tasks (Ju et al., 2023; Wang et al., 2024) and allow our framework to generalize to previously unseen drugs by projecting molecular embeddings into FM's internal state.

By featurizing cell lines based on their initial gene expression, our framework can generalize to unseen cell lines in both zero-shot and few-shot settings. In particular, by using pre-trained weights in the original transformer layers, the model can leverage gene-gene interactions learned through large-scale pre-training (Cui et al., 2024) without the need to explicitly encode them into the model, as done in previous works (Roohani et al., 2024). Importantly, the ability to generalize to new cell lines arises from leveraging knowledge available in the pre-trained model and scDCA preserving and integrating it with molecular representations during finetuning.

## 4 EXPERIMENTS

We show that scDCA achieves superior results for all tasks described in Section 3, while having less than 1% of the number of parameters in the original single cell FM.

### 4.1 EXPERIMENTAL SETTINGS

We evaluate our framework *scDCA* on 4 tasks: (1) unseen drugs, (2) unseen drug-cell-line, (3) unseen cell line (few-shot), and (4) unseen cell line (zero-shot). We compare the proposed approach with several recently introduce methods for perturbation prediction: *chemCPA* (Hetzel et al., 2022), *BioLORD* (Piran et al., 2024), and *Sams_VAE* (Bereket & Karaletsos, 2024).

**Dataset.** The dataset utilized in this study is sciplex3 (Srivatsan et al., 2020), which includes 649,340 cells across 7,561 drug-sensitive genes on 3 human cancer cell lines (A549, MCF7, and K562) perturbed with 188 drugs. Similar to previous work (Hetzel et al., 2022), we focus on the transcriptional response captured through 2,000 drug-sensitive genes.

**Data Splitting.** As previously described in Section 3 and Figure 2, we evaluate scDCA across different generalization tasks. In practice, each task results from a different data splitting strategy, detailed in Appendix A.1.

In this study, as we focus on single-cell data, we observe that the gene expression of only a subset of genes undergoes significant alterations under various perturbations. To ensure a meaningful evaluation, our analysis specifically targets those genes that exhibit noticeable changes in their expression levels compared to their initial (control) expression profiles. These genes are categorized as differentially expressed genes (DEGs). The number of DEGs identified can vary; however, it is generally observed (Roohani et al., 2024) that analyzing a smaller, more defined set of DEGs presents a more robust and relevant challenge, as it requires precise detection of nuanced expression changes. In our experiments, we consider the top 20 DEGs.

**Evaluation Metrics.** Similar to previous works (Hetzel et al., 2022), the main performance metric used for perturbation prediction is the R-squared (or coefficient of determination) metric $R^2$. This metric quantifies the proportion of the variance in the dependent variable that is predictable from the independent variables in a regression model. We conducted all experiments over 5 runs (different random splits), reporting both the standard error and the average coefficient of determination. By focusing the evaluation on the top DEGs, we make sure that the metric is not inflated by genes that do not change significantly after the perturbation compared to control.

## 4.2 RESULTS

We compared different fine-tuning strategies (including scDCA) and recently introduced baselines across all tasks. In the following, we summarize the main findings.

**scDCA outperforms other finetuning approaches**. We first compare the performance of scDCA with the naive finetuning method described in Section 3.2. The results are summarized in Table 1. scDCA consistently outperforms the naive finetuning approach across all tasks. Notably, as the tasks become more challenging (few-shot and zero-shot settings), the performance gap widens. This indicates that while fine-tuning scGPT performs adequately for unseen drug (or drug-cell-line) prediction, leveraging scDCA yields a significant boost in the generalization to novel cell lines. Additionally, scDCA utilizes less than 1% of the parameters required by the naive fine-tuning approach, resulting in a faster and more memory-efficient procedure. In particular, we notice how, for the task of zero-shot cell line prediction, standard fine-tuning leads to significantly lower results, likely due to overfitting to training cell lines, without the ability to preserve original representations for test ones. While we observe scDCA results to be robust across the different tasks, direct comparison of the performance across tasks is challenging, given differences in splitting strategies and the unique features of each task.

| Task | scDCA | Fine-tuning | $\Delta$ |
|---|---|---|---|
| Unseen drug | **0.81 ± 0.022** | **0.81 ± 0.021** | 0.00 |
| Unseen drug-cell-line combo | **0.83 ± 0.015** | 0.78 ± 0.009 | 0.05 |
| Unseen cell line (few-shot) | **0.88 ± 0.004** | 0.81 ± 0.005 | 0.07 |
| Unseen cell line (zero-shot) | **0.82 ± 0.032** | 0.51 ± 0.021 | 0.31 |

Table 1: Comparison of scDCA and scGPT Finetuning (Mean ± SE)

**scDCA outperforms all baselines across tasks.** As summarized in Figure 3, scDCA consistently outperforms the baseline models for all tasks, with the most notable improvement observed in the unseen cell line (zero-shot and few-shot) tasks. scDCA leverages extensive domain knowledge, such as gene co-occurrence, to effectively generalize to new scenarios. This is particularly beneficial in few-shot and zero-shot settings, where minimal data is available. The superior performance of scDCA is further confirmed by its lower variance across data splits, especially for the generalization to new drugs and the few-shot prediction in unseen cell lines. Finally, we observe how ChemCPA is unable to perform the unseen cell line (zero-shot) task due to its architecture, which relies on trained cell line embeddings as a core component, limiting its ability to handle unseen cell lines.

**scDCA predictions are within single-cell measurement uncertainty.** After quantitatively evaluating the performance of different models, we inspected the prediction of individual genes after perturbation compared to control (initial gene expression). Ideally, a model should be able to characterize fine-grained differences in gene expression. However, the noise inherent in the measurement procedure imposes a ceiling on model performance. Figure 4 presents examples of predicted gene expression across 20 DEGs following perturbation for different chemical structures and two tasks. As shown, scDCA successfully captures log-fold changes up to the standard deviation of the underlying single-cell distribution. Results for the other tasks are shown in Appendix (Figure 6), highlighting the same trend.

**scDCA predictions are robust across different targets.** After assessing the model's accuracy, we evaluated its robustness for different perturbagens. In particular, we asked whether the performance of the model is consistent across molecular targets. We utilized the ChEMBL database (Gaulton et al., 2012) to identify known targets for the molecules present in our dataset. This analysis revealed known

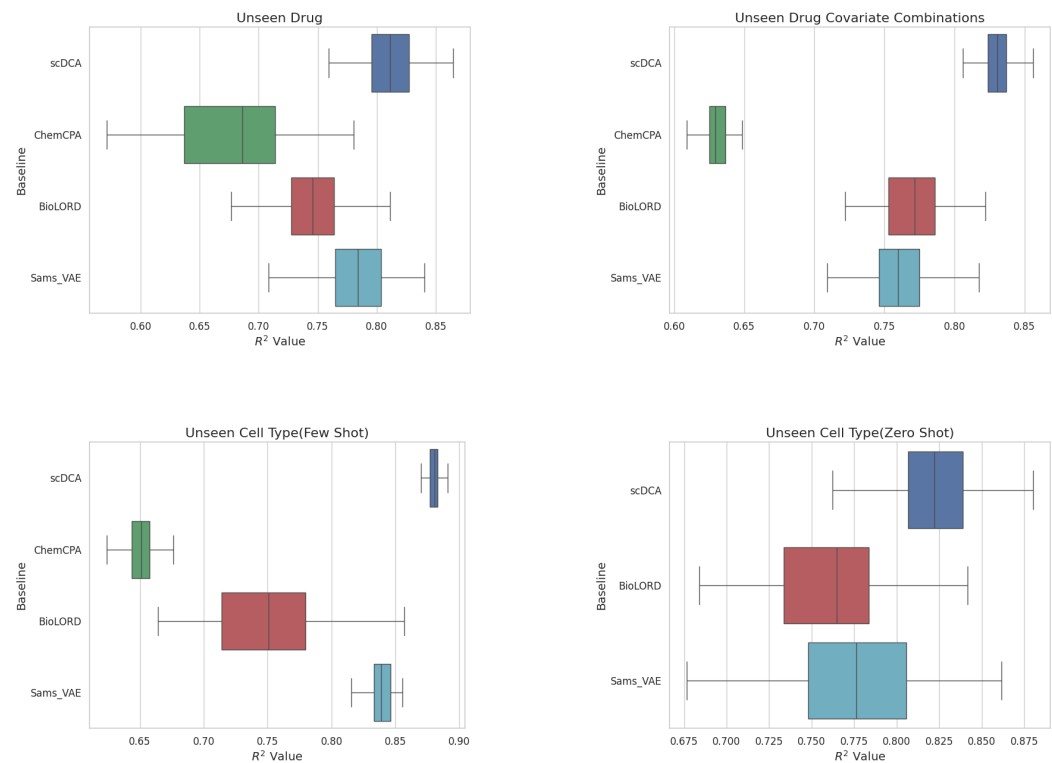

Figure 3: Comparison with different baselines. scDCA is our proposed method. X-axis represents $R^2$ (Mean $\pm$ SE).

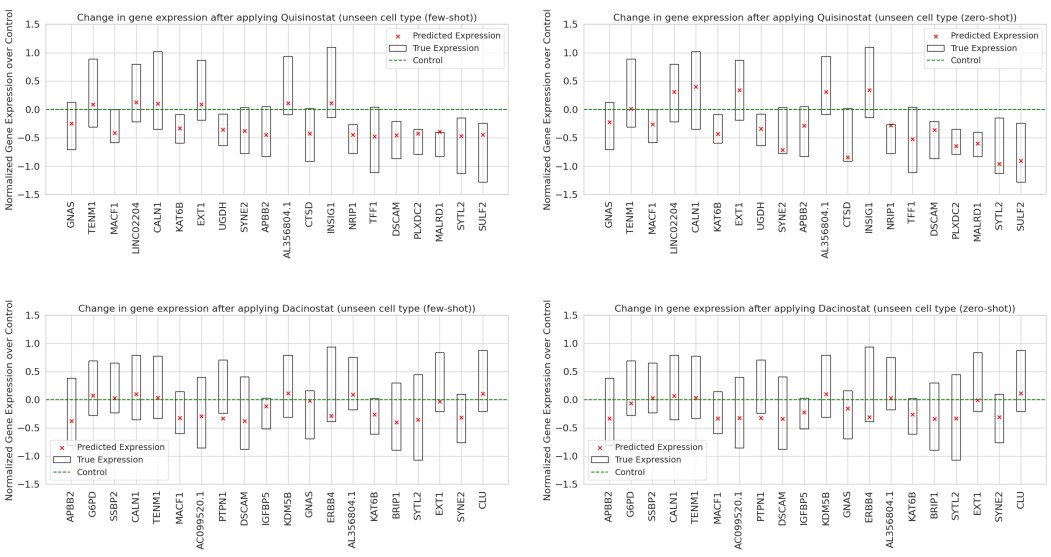

Figure 4: Examples of predicted gene expression across 20 most differentially expressed genes for the molecules Quisinostat and Dacinostat.

targets for 77 molecules. We grouped $R^2$ for each target-cluster, aggregating the performance on the relative subset of molecules. Figure 5 summarizes this experiment. As shown, scDCA's predictive

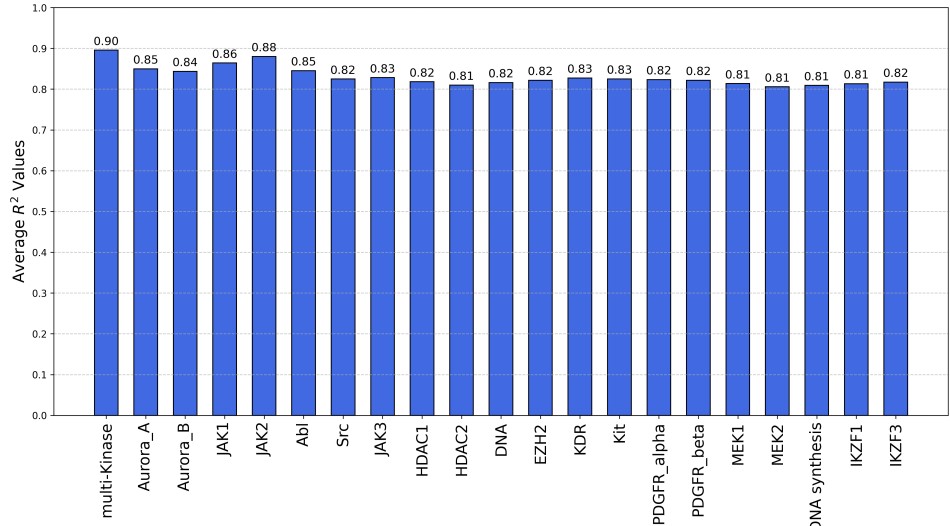

Figure 5: Clustering drugs based on their targets. X-axis represents targets.

accuracy is consistent across targets, with no significant drops in performance for any subgroup. More details of this experiment are available in Section A.6.

## 5 CONCLUSION AND LIMITATIONS

In this paper, we address the underexplored area of predicting transcriptional cellular responses to novel molecular perturbations. A key challenge in this area is data scarcity, a problem that foundation models trained on much larger ancillary datasets are poised to address. While naive applications of FMs fall short of this expectation, we demonstrate that a careful fine-tuning strategy combining a single-cell FM with a pre-trained molecular FM indeed realizes this potential. Our strategy, single-cell drug-conditional adapter (scDCA), leads to state-of-the-art performance on a variety of challenging problem formulations, including the few-shot and zero-shot generalization to unseen cell lines.

One limitation is that our approach is currently only applicable to transformer-based foundation models. This restriction means that (scDCA) cannot be directly applied to other types of models without significant adaptation. Additionally, our method requires the presence of control gene expression data. If the dataset does not include control gene expressions, our approach cannot be utilized. Finally, we believe that the evaluation of perturbation models, especially in the single-cell domain and for the generalization to new cell lines and conditions, is challenged by the limited size of publicly available data. Despite these limitations, we believe our approach offers valuable insights and advancements in the field of chemical perturbation prediction. Future work will aim to tackle these challenges.

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

# A APPENDIX

## A.1 EXPERIMENTAL SETTING AND DATA SPLITTING

In this paper, we evaluate the performance of our model and baselines under the following conditions: the number of epochs is set to 20, the batch size is 16, and the number of runs is 5. We use similar hyperparameters as the original implementation of all the baselines introduced in their papers.

For the task of unseen drugs, we partitioned the molecules into two distinct sets for training and testing with a ratio of 70% training data and 30% testing data. For the task of unseen drug-cell-line combination, we partition the data into training and testing sets with a ratio of 50% training data, 50% testing data sampled uniformly at random, so that the training set may include data where the compound has been observed in different cell lines. For the task of unseen cell lines (zero-shot), we reserved one cell line entirely for testing (except the negative control measurement) and trained the model on the remaining cell lines. Finally, for the task of unseen cell lines (few-shot), similar to the zero-shot task, we trained on two cell lines but split the data on the remaining cell line into 10% training, 90% testing data.

## A.2 DETAILS OF THE SCDCA MODEL AND TRAINING PROCEDURE

**Preprocessing.** For each combination of cell line $c$ and drug $d$, we calculate the average expression vector $X^{(d)}(c)$ and denote the control expression vector as $X^{(0)}(d)$. This average control gene expression vector, along with gene tokens, is used as input to our model.

**Data splits.** We first split our data into training and test sets based on four different tasks. The different tasks/splits are illustrated in Figure 2.

**Inputs.**

- Control cells expression vector $X^{(0)}(c)$ for cell line $c$,

- Molecular structure $d$.

**Prediction target.** Expression vector $X^{(d)}(c)$ for drug $d$ in cell line $c$.

**Frozen parameters/networks.**

- Initial gene tokens $\text{emb}_g(t_g)$,

- Gene expression embedding $\text{emb}_X$,

- Molecule embedding $\text{emb}_m$ (ChemBERTa),

- For every layer $l = 1, \ldots, L$, scGPT transformer layer $\text{scGPT}_l$.

**Trainable parameters/networks.** For every layer $l = 1, \ldots, L$, we have the following trainable parameters of our adapter layer:

- Molecular embedding projections $f_l^m$,

- Projection matrices $W_l^{\text{down}}, W_l^{\text{up}}$,

- residual network parameters $\text{res\_net}_l$.

**Model architecture.** First, the initial gene-level embedding tokens are calculated as:

$$h_0 = \{h_0^{(g)}\}_g = \{\text{emb}_g(t_g) + \text{emb}_X(X^{(0)}(c))\}_g, \tag{6}$$

where here and in the following, all operations on $h_l$ are to be understood to operate over the whole set of genes $g$ for notational convenience.

Next, $h_0$ gets passed through the scGPT transformer layers with the additional molecular adapters (Figure 1). That is, for $l = 1, \ldots, L$,

$$
\begin{align}
h_l &= \text{scGPT}_l(h_{l-1}), & \text{(scGPT embedding)} && (7) \\
b_l &= f_l^m(\text{emb}_m(d)), & \text{(Molecular embedding projection )} && (8) \\
h_l^{\text{down}} &= W_l^{\text{down}} h_l + \Pi^{\text{down}} b_l, & \text{(Feed-forward down-project)} && (9) \\
h_l^{\text{res\_net}} &= \text{res\_net}_l(h_l^{\text{down}}), & \text{(Residual network)} && (10) \\
h_{l+1} &= h_l^{\text{up}} = W_l^{\text{up}} h_l^{\text{res\_net}} + b_l. & \text{(Feed-forward up-project)} && (11)
\end{align}
$$

In our model, we use $L = 12$ layers.

**Loss function.** Finally, the loss function is calculated as the mean squared error loss over genes,

$$
\mathcal{L} = \frac{1}{G} \sum_{g=1}^{G} \left( \text{MLP}(h_L^{(g)}) - X^{(d)}(c) \right)^2. \tag{12}
$$

**Model training.**

- **Optimizer**: The model is trained using the Adam optimizer.
- **Learning rate**: We use a learning rate of 1e-4.
- **Batch Size**: The batch size used during training is 16.
- **Epochs**: The model is trained for 20 epochs.

## A.3 ABLATION STUDIES

We compare scDCA with two alternative models. In the first model, we replace the res_net layer with a non-linearity module (similar to the original adapter paper Houlsby et al. (2019)). For the second model, we concatenate the molecule embedding with the output of scGPT to predict the effect on gene expression. These results are summarized in Table 2.

Table 2: Ablation study. We compare scDCA with two other models: (1) instead of res_net, we use a nonlinearity in the original adapter paperr Houlsby et al. (2019). (2) We concatenate the molecule embedding with the output of scGPT to predict the effect on gene expression.

| Task | scDCA | scDCA w/o res_net | scGPT+mol_emb |
|------|-------|-------------------|---------------|
| Unseen Drug-Cell-Line Combinations | **0.83** | 0.81 | 0.80 |
| Unseen Drugs | **0.82** | 0.81 | 0.78 |
| Unseen Cell Line (Few Shot) | **0.88** | 0.87 | 0.72 |
| Unseen Cell Line (Zero Shot) | **0.82** | 0.80 | 0.31 |

## A.4 FEATURE-BASED FINE-TUNING VS SCDCA

In Section 3.2 we mention that a common way to fine-tune foundation models is through feature-based fine-tuning. In Table 3, we demonstrate that this approach yields poor results when compared to our proposed method.

| **Task** | **scDCA** | **Feature-based** | $\Delta$ |
|----------|-----------|-------------------|----------|
| Unseen Drug | **0.81 $\pm$ 0.022** | 0.03 $\pm$ 0.011 | 0.78 |
| Unseen Drug-Cell-Line Combination | **0.83 $\pm$ 0.015** | 0.01 $\pm$ 0.019 | 0.82 |
| Unseen Cell Line (Few Shot) | **0.88 $\pm$ 0.004** | 0.01 $\pm$ 0.025 | 0.87 |
| Unseen Cell Line (Zero Shot) | **0.82 $\pm$ 0.032** | -1.51 $\pm$ 0.011 | 0.82 |

Table 3: Comparison of scDCA and feature based approach (Mean $\pm$ SE)

## A.5 ADDITIONAL EXAMPLES OF SCDCA PREDICTIONS

We show examples of predicted gene expression across 20 DEGs following perturbation for different chemical structures (similar to Figure 4) for the tasks of unseen drug and unseen drug-cell-line combination prediction in Figure 6.

## A.6 ADDITIONAL DISCUSSION ON THE ROBUSTNESS OF SCDCA

The design of the experiment for Figure 5 in the main text is as follows. First, we obtain compound target annotations from ChEMBL. We then conduct a compound hold-out (unseen drug) experiment, ensuring that all annotated compounds are reserved for the test set. For evaluation, we subset the test set compounds to one target cluster at a time. We then report the R2 values for each target cluster.

### A.6.1 COMPARISON OF THE ROBUSTNESS OF SCDCA AND CHEMCPA ACROSS DIFFERENT CLUSTER-TARGETS

We compare the robustness of scDCA across different targets (shown in Figure 5) with ChemCPA in Figure 7.

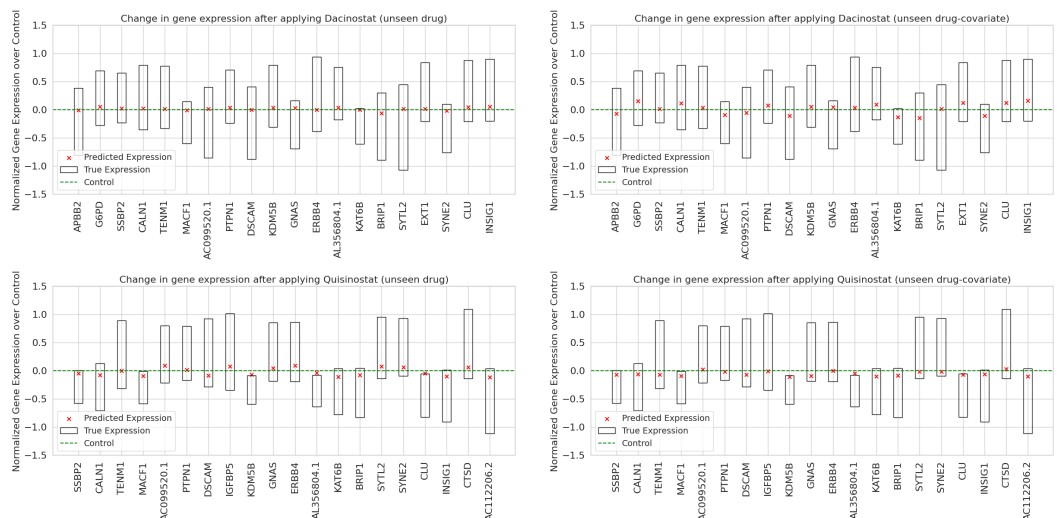

Figure 6: Examples of predicted gene expression across 20 most differentially expressed genes for the molecules Quisinostat and Dacinostat.

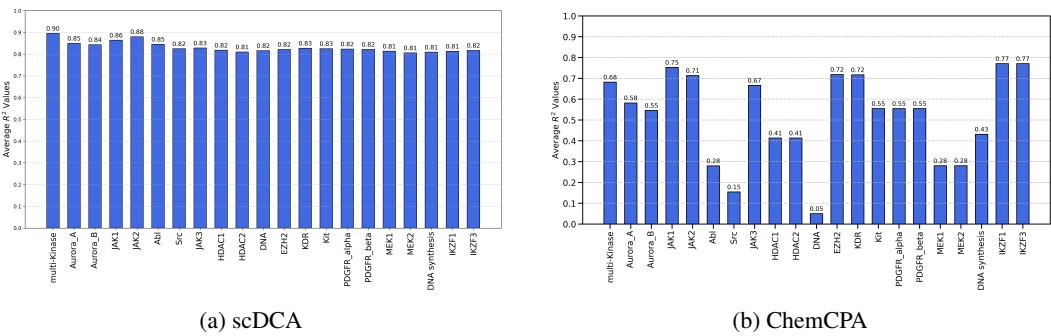

(a) scDCA

(b) ChemCPA

Figure 7: Clustering drugs based on their targets. X-axis represents targets.

## A.7 PERFORMANCE OF SCDCA ON VARIOUS SIZES OF DIFFERENTIALLY EXPRESSED GENES (DEGS)

We further evaluate the performance of scDCA across varying sizes of differentially expressed genes (DEGs). In the main paper, we show our results for the top 20 differentially expressed genes. Here we show how the performance changes as we change the size of our DEGs. We expect the performance of the model to improve with a larger set of genes, as many genes are not significantly affected by the perturbation. However, focusing on a small set of DEGs represents a more challenging and relevant task, as it shows the ability of a model to identify the most significant changes. Figure 8 summarizes these results.

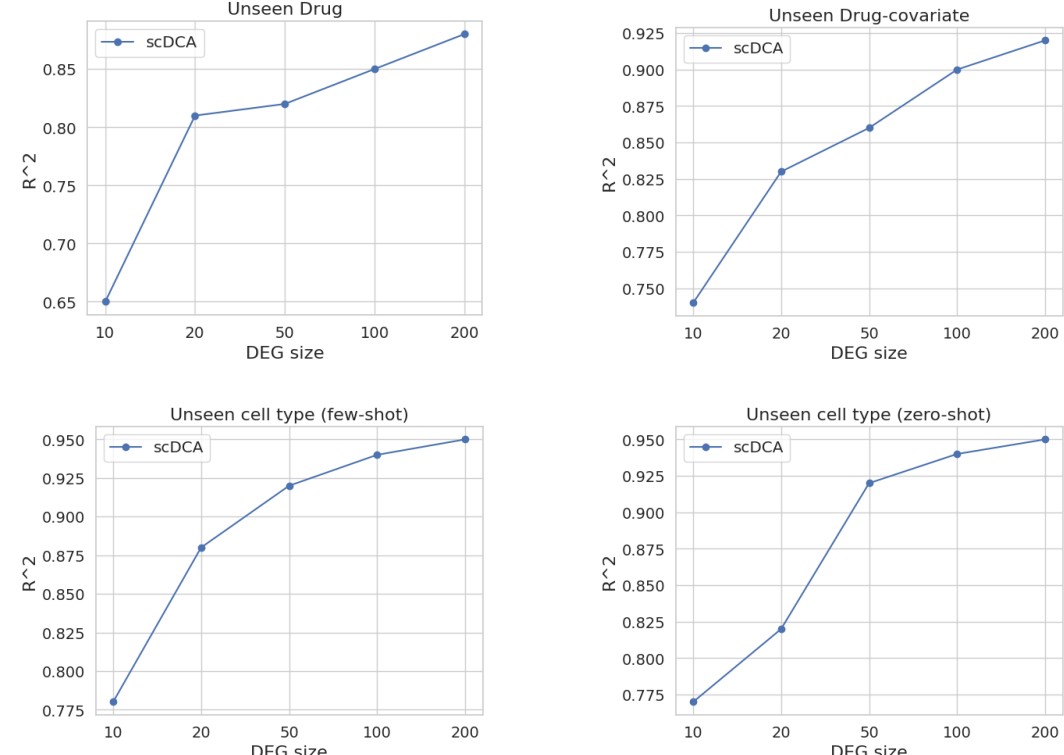

Figure 8: $R^2$ values for different tasks on different numbers of DEGs .

### A.8 PAIRED T-TESTS RESULTS OF SCDCA VERSUS THE OTHER MODELS

We performed paired t-tests to compare the performance of scDCA against ChemCPA, BioLORD, and SAMS_VAE across multiple tasks. The results are summarized in Tables 4, 5, 6, and 7.

For all tasks, the scDCA demonstrated a statistically significant improvement over all baselines.

| Comparison | T-statistic | P-value | Significant (p < 0.05) |
|---|---|---|---|
| scDCA vs ChemCPA | 4.5 | 0.01 | Yes |
| scDCA vs BioLORD | 3.3 | 0.03 | Yes |
| scDCA vs SAMS_VAE | 4.3 | 0.01 | Yes |

Table 4: Summary of paired t-test results for the task of unseen drug.

| Comparison | T-statistic | P-value | Significant (p < 0.05) |
|---|---|---|---|
| scDCA vs ChemCPA | 9.0 | 0.003 | Yes |
| scDCA vs BioLORD | 7.3 | 0.005 | Yes |
| scDCA vs SAMS_VAE | 10.7 | 0.002 | Yes |

Table 5: Summary of paired t-test results for the task of unseen drug-cell-line combination.

### A.9 GENERALIZATION TO NEW CELL LINES AND SIMILARITY BETWEEN CELL LINES

Since there are only three cell lines present in our dataset, we trained our model and baselines in a leave-one-out manner for cell line generalization tasks. In general, we would expect the difficulty of the unseen cell line task to vary according to the degree of biological similarity of the cell lines in

| Comparison | T-statistic | P-value | Significant (p < 0.05) |
|---|---|---|---|
| scDCA vs ChemCPA | 31.5 | 0.000 | Yes |
| scDCA vs BioLORD | 3.08 | 0.027 | Yes |
| scDCA vs SAMS_VAE | 6.33 | 0.003 | Yes |

Table 6: Summary of paired t-test results for the task of unseen cell line (few shot)

| Comparison | T-statistic | P-value | Significant (p < 0.05) |
|---|---|---|---|
| scDCA vs BioLORD | 3.7 | 0.01 | Yes |
| scDCA vs SAMS_VAE | 3.9 | 0.01 | Yes |

Table 7: Summary of paired t-test results for the task of unseen cell line (zero shot)

training and test. Table 8 summarizes the results for each split (i.e., for each test cell line) for the task of unseen cell line (zero shot).

Two of the considered cell lines are adenocarcinoma cell lines, albeit in different tissues, while the third one is a leukemia cell line. Because of this, one might expect the leukemia cell line (K562) hold out to be the hardest split, which is indeed the case for scDCA. However, this trend is not present in BioLORD and SAMS_VAE, the other two methods under consideration.

| Held out cell line | scDCA | BioLORD | SAMS_VAE |
|---|---|---|---|
| A549 (lung adenocarcinoma) | **0.81** | 0.71 | 0.73 |
| K562 (myelogenous leukemia) | **0.77** | 0.74 | 0.75 |
| MCF7 (breast adenocarcinoma) | **0.88** | 0.84 | 0.84 |

Table 8: Summary of zero-shot prediction for unseen cell lines.

## A.10 SIMILARITY OF DRUGS IN TRAINING AND TEST SETS GENERALIZING TO UNSEEN DRUGS

We analyzed the similarity between training and test molecules, focusing on the generalization to new molecules (drugs). For each test molecule, we computed its maximum similarity to any training molecule. We used standard functions, leveraging Tanimoto similarity on Morgan fingerprints (radius=2, num_bits=2048) as implemented in RDKit library. Figure 9 summarizes these results. As shown, the vast majority of molecules have low similarity to training structures, with 88% of them having a similarity 0.4 to any training molecule (0.4 is often considered the threshold to define structural novelty) (Dalke, 2019).

## A.11 TRAINING TIME AND MEMORY COMPARISON

Our model was trained on a single NVIDIA L40S GPU. Table 9 reports the training time (per epoch) and the memory requirement for scDCA, also compared to full model fine-tuning.

| Model | Training time (s) | Memory (GB) |
|---|---|---|
| scDCA | 1.07 | 45 |
| Fine-tuning | 1.20 | 53 |

Table 9: Comparing scDCA running time and memory usage with fine-tuning the full model.

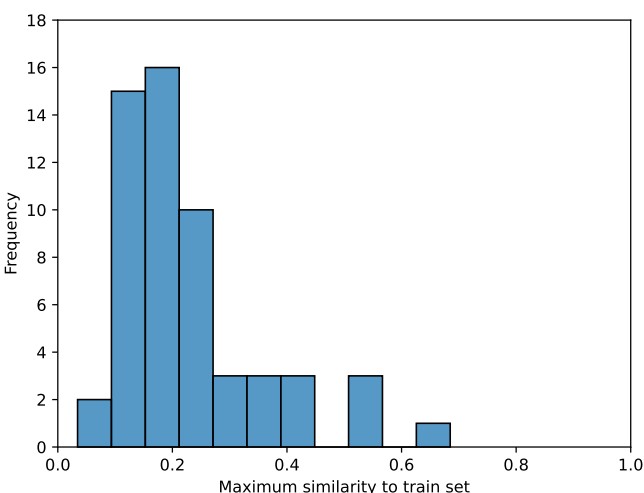

Figure 9: Similarity of unseen drugs. Histogram of maximum Tanimoto similarity to the training set for one train/test split in the unseen drug task.

