# OpenReview forum: "Efficient Fine-Tuning of Single-Cell Foundation Models Enables Zero-Shot Molecular Perturbation Prediction"
_ICLR.cc/2025/Conference — Submitted to ICLR 2025_

### Official Review · Reviewer_jwc9 · 2024-11-02

**Soundness:** 3
**Presentation:** 3
**Contribution:** 3
**Rating:** 6
**Confidence:** 3

**Summary:**

This paper introduces a parameter-efficient approach to fine-tune single-cell foundation models for predicting how cells respond to drug treatments. They develop a "drug-conditional adapter" that allows them to inject information about molecular structures (from external embeddings) into the model while keeping the original foundation model frozen. Their method enables both prediction of cellular responses to new drugs and generalization to entirely new cell types that weren't seen during training. They demonstrate their approach outperforms existing methods particularly in few-shot and zero-shot scenarios where the model must generalize to new cell types.

**Strengths:**

- practical approach: leveraging foundation models for drug response prediction is highly relevant for drug discovery
- Interesting approach to solving the challenge of incorporating molecular structure information into gene expression models
- Comprehensive evaluation framework testing multiple important scenarios (unseen drugs, drug-covariate pairs, zero-shot). Generally I find the results fairly convincing.
- Results on zero-shot generalization to new cell types are valuable for real-world applications

**Weaknesses:**

- I find the rationale for the adapter a bit unconcinving in it's current state:
	- the authors talk about parameter efficiency but the base model is only 53M parameters so can easily and quickly be finetuned on a consumer GPU. As such, I don't find the parameter efficiency a good selling point of this paper.
	- I'm surprised how poorly the full model finetune performs in some of the settings. Probably I'd find the paper even more useful if the authors managed to get full model FT working, since that's easier to replicate (less complexity).
- The rationale behind the bottleneck design isn't well explained
- Missing important experimental details that would help reproducibility (GPU requirements (compared to full finetune), training time, hyperparameter selection). Could you add these to the paper?
- Several broken citations need fixing (their venue is not listed): Chemberta, Bert and a few others.
- The statement about residual connections facilitating drug-cell type interactions lacks citation / grounding (line 331)

**Questions:**

1. The ResNet component in their adapter architecture is inadequately explained. When I hear ResNet, I think about ResNet the model whereas the authors  seem to be using it to mean a generic residual connection. Could you clarify exactly what's happening in the ResNet block of your adapter?
2. Why did you choose a bottleneck design for the adapter rather than a larger architecture? What were the key considerations?
3. Could you provide more details about computational requirements (GPU usage, training time, memory requirements) for both your adapter approach and the full fine-tuning baseline?
4. Figure 4 lacks labels. I assume these are the same order as table 1?

---

> ### Author Response · Authors · 2024-11-21
> **Rebuttal to Reviewers' Comments (part 1)**
>
> Thank you for your detailed comments. We truly appreciate that you found the proposed approach practical and interesting, with convincing results.
>
> We have addressed all the points raised below, including providing additional analysis, figures, and edits to the manuscript.
>
> **The rationale for the adapter and the efficiency**
>
> Thank you for your observation. While parameter efficiency can generally improve memory and computing requirements, in our setting, we are primarily focused on supporting data efficiency.
>
> Predicting outcomes of molecular perturbations is critical in drug discovery; however, it is challenged by very limited data available for this task, making it a few-shot (or zero-shot) problem in many important settings. For example, the sciplex3 dataset used in this study (and previous studies) includes only 3 cell types and 188 drugs.
> It is true that the original model has 53M parameters. However, due to the limited data available for this task, it is not ideal to fine-tune all these parameters, as it can be very easy to overfit the small dataset and forget the important biological knowledge learned during pre-training. Therefore, the need for a parameter-efficient approach in our study primarily allows fine-tuning on small chemical perturbation datasets, avoiding overfitting while preserving important background biological knowledge encoded in the pre-trained weights. This approach helps us improve generalization performance, especially in few-shot and zero-shot settings.
>
> To better clarify this point, we added the following sentence in the revised introduction: “Our parameter-efficient approach allows fine-tuning on small chemical perturbation datasets, avoiding overfitting while preserving important background biological knowledge encoded in the pre-trained weights.”
>
> **Performance of full model finetuning**
>
> Thank you for your feedback. As mentioned in the previous point, we believe the main reason the full model fine-tuning performs worse is that it involves fine-tuning a large number of parameters on a very small amount of data. Fine-tuning such a large model with limited data can lead to overfitting and suboptimal performance, especially in few-shot and zero-shot generalization tasks.
> As we have shown in Table 1, full model finetuning indeed achieves comparable results to scDCA in the unseen drug setting and good results in the unseen drug-cell-line combo setting. However, the results of full model finetuning are significantly worse in the unseen cell line (few-shot and zero-shot) settings. We attribute the improved results of scDCA in such challenging settings to the ability to retain biological information about unseen cell lines during finetuning.
>
> **The rationale behind the bottleneck design isn't well explained**
>
> The rationale behind the bottleneck design is similar to that of previous adapter-based finetuning papers [1]. Essentially, this design allows us to significantly reduce the number of parameters by compressing the original embedding size to a much smaller one.
>
> [1] Houlsby et al., Parameter-Efficient Transfer Learning for NLP, NeurIPS (2019).
>
>
> **Missing important experimental details**
>
> Thank you for your helpful comment. Following the suggestion, we have added extensive details on the scDCA model architecture and training procedure, including hyperparameters, in Section A.2 in the Appendix.  We believe that these details, combined with the code release, will ensure the full reproducibility of the work. Furthermore, we have added training time and memory usage in Section A.11 in the Appendix. As shown, scDCA has reduced training time and memory requirements compared to full model fine-tuning (even though, as previously mentioned, this is not the main goal of the method).
>
> **Several broken citations need fixing**
>
> Thank you for your observation. We have tried to fix all those that we could find.
>
> **The statement about residual connections facilitating drug-cell type interactions lacks citation**
>
> Thank you for your helpful suggestion. What we meant with that statement is that residual connections were useful to (marginally) improve model performance. We recognize the sentence was inaccurate and confusing, therefore:
> - We removed that statement from the main text, and
> - We added an ablation study to Section A.3 in the Appendix to show the effect of the residual network in our architecture. The result from this experiment demonstrates an improvement when using residual connections.

---

> > ### Author Response · Authors · 2024-11-21
> > **Rebuttal to Reviewers' Comments (part 2)**
> >
> > **What's happening in the ResNet block of your adapter?**
> >
> > Thank you for your feedback. Indeed, we are referring to a generic residual network in this context. We have clarified this term and now refer to it as “res_net” throughout the manuscript to avoid confusion. The res_net block in our implementation is a residual network that consists of a two-layer MLP with ReLU activation function and layer normalization applied after each layer. A skip connection is used to combine the input and output of the residual network.
> >
> > **Figure 4 lacks labels**
> >
> > Thank you for your observation. We have fixed this in the revised version.

---

> ### Comment · Reviewer_jwc9 · 2024-11-27
>
> I think the authors for taking the time to respond to my review. I have gone through the paper one more time and while I still believe the paper should be accepted, I have decided to not raise my score. The fine tuning approach strikes me as overly complex, and I do not find the author's justification of overfitting avoidance convincing. Overcomplicated modeling lowers the scientific contribution of this work since it makes it hard to use on novel task as well as replicate. Since the evaluation is restricted to a single dataset I do not find the complexity justified.

---

> > ### Author Response · Authors · 2024-12-04
> >
> > We would like to sincerely thank you for the valuable and supportive feedback and for supporting the acceptance of this work.
> >
> > We agree that exploring other lightweight approaches besides what we have investigated in our work (full-finetuning or feature-based training) would be an interesting direction for future research that we aim to pursue, especially as more and larger public datasets become available.

---

### Official Review · Reviewer_Le4o · 2024-11-03

**Soundness:** 2
**Presentation:** 1
**Contribution:** 2
**Rating:** 3
**Confidence:** 5

**Summary:**

The paper's authors introduce scDCA, a fine-tuning strategy for foundation models to predict single-cell RNA-seq perturbations in a zero-shot fashion. A pre-trained transformer model (in this case, scGPT) is adapted to the task of perturbation prediction by introducing an adapter module. The authors freeze most of the foundation model and only train the adapter module downstream. Such a module inputs drug representations (different modalities) and uses them to steer average control expression to its drug-perturbed counterpart. Moreover, scDCA can deal with unseen cell types by using their average control expression as input. In this way, scDCA is flexible: It can tackle missing compounds, cell types and combinations thereof. The authors present superior performance in comparison with four perturbation models on the task of recapitulating the perturbation effects based on differentially expressed genes. Furthermore, the paper presents qualitative results of successful drug effect prediction by showing that the model induces a gene expression shift on control compatible with the real perturbation direction.

**Strengths:**

I think the idea at the core of the paper is really original, interesting and promising. With the increase in the size of modern single cell datasets and representation models, I believe that research should focus on how to build additional ML tasks as efficiently as possible. The paper is therefore timely and interesting in the modern context. An additional positive aspect is that the performance appears to be good and solid according to the results provided by the authors.

**Weaknesses:**

* In the introduction a sentence is formulated as follows “Additionally, several methods focus on predicting the effects of genetic perturbations, where the number of possible treatments is fixed (approximately 20,000 genes) (Roohani et al., 2023). These approaches are not directly applicable to explore the vast chemical space, which is estimated to encompass up to 10^60 drug-like compounds”. In the first sentence, you talk about genetic perturbations, but then the vastness of the chemical space is brought up. I found this a bit confusing.
* In my opinion, it is a little bit unfair to define the model as “state of the art”. The proposed evaluation is not exactly the same as in e.g., chemCPA or CPA. Specifically, the model’s performance has not been tested on held-out compounds like in the aforementioned models. I believe the authors should maybe add a disclaimer on the level of superiority their model implies.
* There is a bit of an overuse of the variable c. First, it is introduced as a cell type indicator, while later it is used as a gene-specific attribute in the description of scGPT. I think the presentation would be clearer with a different notation.
* Line 261-262: since it is a claim, it would be nice to have some citations!
* When presenting the feature-based approach in 3.2, it is not clear what kind of model you compare the scDCA approach within Appendix A.2. Since you are doing perturbation prediction, do
* In Eq. 4, the authors introduce a control embedder. Connecting it to what is explained in line 10, the authors take the average embeddings of the controls of a cell type c and perturb it synthetically using the model. However, if I understand correctly, the value $X^0_c$ is an average. Does it mean that only the average control is subjected to synthetic perturbation? Are you not neglecting a lot of information on potential control heterogeneity?
* I think the paper lacks a more rigorous description of the training process that can be integrated into the extra page. I would encourage the authors to explicitly define the target of the perturbation prediction task when starting from the average control expression. Probably the presence of an algorithm would help. Currently, I struggle to understand what task is used to ensure plausible predictions by the FM.
* In my opinion, Figure 3 is really hard to understand. I am not sure which plot refers to which task.
* Is the benchmark against existing models performed on held-out genes? This is not mentioned in the text describing the experiments. In the negative case, I would definitely expect to see a comparison in performance against chemCPA on leave-out compound predictions, using the same splits as in the original paper.
* Figure 4 does not convince me completely. I would find it beneficial to, for each gene, overlay the control distribution on the perturbed distribution to confirm that the model is really predicting extreme values of the control distribution.
* I think the conclusion should also contain some limitation discussion.
* In my opinion, the appendix is not well-curated and organized. Plot labels are cut too. I do understand that the may be a matter of time, but I would encourage the authors to increase the amount of detail in future iterations of the work, especially on the baseline comparison experiments.
* To appreciate the value of Figure 5 I think I would need to see a comparison with other perturbation models like chemCPA.

Although potentially promising, I am not sure the paper is ready for publication yet. In my opinion, the presentation of the methods should be more thorough and detailed. I am happy to discuss my doubts with the authors during rebuttals and potentially change my opinion.

**Questions:**

* Does the adapter fit well with other types of foundation models aside from scGPT? I think it would be great to introduce an Appendix discussion about this.
* In Sec 3.1, lines 209-214 I struggle to understand the logic. In the first sentence, it is claimed that control profiles are aggregated as cell line representations. Then you go on saying that you do it for all cell line drug pairs. What are the drug-cell line covariate embeddings used for? I would really appreciate it if the authors could elaborate on this a bit more and clarify this point.
* To me, it is not very clear what objective is used in 3.2. You input the (average) control expression, the condition and the identity token. Then what loss do you calculate to evaluate the correctness of the prediction? You mention the loss in Equation 3, but how is it applied here to make sure an average control expression Is correctly perturbed? To me, this aspect is a bit unclear across the whole paper.
* Is the evaluation carried out to the 20 top DEG or also the prediction? I.e. Do you both predict and evaluate on a limited number of genes or predict on the whole transcriptome and evaluate on the subset of genes?
* Are all baselines able to perform all tasks out of the box? Or do you have to adapt them for them? I think it would be beneficial to include a description of of baseline models and how they were run for comparison in the appendix.

---

> ### Author Response · Authors · 2024-11-20
> **Rebuttal to Reviewers' Comments (part 1)**
>
> Thank you for your detailed comments. We truly appreciate that you found the proposed approach original, promising, and relevant.
> We have addressed all the points raised below, including providing additional analysis, figures, and edits to the manuscript.
>
> **“In the introduction a sentence is formulated as follows…”**
>
> Thank you for the comment. Our goal with this sentence was to compare the domain of genetic perturbations, where the “alphabet” of possible perturbation is fixed as the set of genes (with their combinations), to the domain of chemical perturbations, where the set of perturbations is the set of (virtually infinite) compounds. The reason for this comparison is that several methods developed for genetic perturbations (e.g., Roohani et al., 2023) are not directly applicable to the prediction of chemical perturbations, because they make genetic-specific assumptions (such as the perturbation relationship graph used in Roohani et al., 2023).
>
> Since this comparison early in the introduction led to some confusion, we now defer this discussion to Section 2 (Related Work).The rewritten paragraph now reads as follows:
>
> “Several methods focus on predicting the effects of genetic perturbations, where perturbagens correspond to genes and their combinations (e.g., Roohani et al., 2023). These approaches are not always directly applicable to the prediction of chemical perturbations, which include the vast chemical space (estimated to encompass up to $10^60$ drug-like compounds (Bohacek et al., 1996)”.
>
> **Model’s performance has not been tested on held-out compounds.**
>
> In fact, one of the different settings that we consider specifically tests the model held-out compounds. In more detail, we would like to highlight that we have tested the proposed model across four different settings. These include (1) unseen drug (corresponding to held-out compounds), (2) unseen drug-covariate [renamed unseen drug-cell-line in the revised manuscript], (3) unseen cell type (few-shot), and (4) unseen cell-type (zero-shot). Figure 2 summarizes the different settings. The first two settings are exactly the same as those used in chemCPA. Specifically, in the unseen drug experiment, we held out 30% of the compounds for testing and trained on the remaining compounds.
> We have improved the illustration in Figure 2 to further clarify the different experiments conducted in the paper.
>
> **Overuse of the variable c.**
>
> Thank you for bringing this to our attention. We have fixed this in the manuscript by replacing the "c" that refers to the gene-specific attributes in the description of scGPT with "cond".
>
> **“Line 261-262: since it is a claim, it would be nice to have some citations”**
>
> Thank you for the comment. Following the suggestion, we have included references across multiple domains: language [1], multimodal modeling [2], as well as applied domains of molecular property prediction [3], and medical images [4].
>
> [1] Wei et al., Finetuned Language Models are Zero-Shot Learners. ICLR 2022.
>
> [2] Shen et al., How Much Can CLIP Benefit Vision-and-Language Tasks?. ICLR 2022
>
> [3] Wang et al., Molecular contrastive learning of representations via graph neural networks. Nature Machine Intelligence. 2022
>
> [4] Zhang et al., A generalist vision–language foundation model for diverse biomedical tasks. Nature Medicine. 2024.
>
> **“When presenting the feature-based approach in 3.2, it is not clear what kind of model you compare the scDCA approach within Appendix A.2. Since you are doing perturbation prediction, do”**
>
> It seems that the rest of your question is incomplete. Could you please provide the complete question so that we can address it accurately?
> To provide some initial clarification, in Section 3.2, we mention that a common way to fine-tune foundation models is through feature-based fine-tuning. In A.4 (previously  A.2), we demonstrate that this approach yields inferior results when compared to our proposed method. We clarified this in the revised version.
>
> **Equation 4 and the average embeddings of the controls**
>
> This is correct, $X_c^0$ is indeed an average. That is, we provide the average of the control embeddings to scDCA for the results presented in the paper. This approach is simpler and more computationally efficient than considering the full distribution. While an alternative method that accounts for the heterogeneity of the control data could potentially capture more nuances, it might also introduce additional noise into the model. Given that our method already yields the best results across tasks compared to baselines, we did not observe a significant need to modify this aspect of our approach.

---

> > ### Author Response · Authors · 2024-11-20
> > **Rebuttal to Reviewers' Comments (part 2)**
> >
> > **I think the paper lacks a more rigorous description of the training process**
> >
> > Thank you for your feedback. We have added Appendix Section A.2 to provide a more detailed description. This new section summarizes the preprocessing, data splits, inputs to the model, prediction target, frozen and trainable parameters, model architecture, loss function, and details of the training procedure.
> >
> > **Clarifying Figure 3**
> >
> > Thank you for your input. We have fixed this in the paper by adding a title to each subpanel in Figure 3.
> >
> > **“Is the benchmark against existing models performed on held-out genes? I would definitely expect to see a comparison against chemCPA…”**
> >
> > Thank you for your question. The benchmark is not performed on held-out genes. We have evaluated the proposed methods across four different settings (see Figure 2 for an overview). These include (1) unseen drug, (2) unseen drug-covariate  [renamed unseen drug-cell-line in the revised manuscript], (3) unseen cell type (few-shot), and (4) unseen cell-type (zero-shot). While the first two settings are shared with chemCPA paper, we introduced (3) and (4) in this manuscript to fully leverage the potential of finetuning pretrained FMs.
> >
> > In our experiments, we do not use the exact same splits as chemCPA; however, we ensure consistency by using the exact same data and splits across all methods in our study.
> >
> > In particular, for unseen drug and unseen drug-covariate settings, we were unfortunately unable to locate the exact splits used by chemCPA, as they are not clearly described in the paper or available in the related repository. However, we have used the same splitting strategies described in the chemCPA paper, and we have made every effort to maintain methodological rigor and consistency in our comparisons.
> >
> > **Figure 4 and control results**
> >
> > Thank you for your question. In Fig 4, we normalize the data against the control, i.e., control results are effectively represented as “zero” in our figures (dashed line). Predicted expression and true expression are shown in comparison to the control. By doing so, we are inherently demonstrating the control expression and how the perturbed distributions deviate from it. For this figure, we followed the same strategy as in Roohani et al., Nat Biotechnol 2024 (GEARS), Fig 2b.
> >
> > **Adding limitation discussion to conclusion**
> >
> > Thank you for your suggestion. One limitation of our work is that our method is currently only applicable to transformer-based foundation models. While this architecture is adopted by the majority of existing single-cell FMs, future work should investigate the extension to different frameworks. Additionally, our method requires the presence of control gene expression data, and it cannot be applied if such information is not available. Finally, we believe that the evaluation of perturbation models, especially for the generalization to new cell types and conditions, is challenged by the limited size of publicly available data, especially in terms of different conditions.
> >
> > We included these points in the conclusion.
> >
> > **In my opinion, the appendix is not well-curated and organized.**
> >
> > Thank you for your feedback. We have taken your comments and suggestions into account and have made significant improvements to the appendix. Specifically, we have reorganized it for better clarity and coherence, improving the figures. Additionally, we have added more experiments and detailed explanations to address the points you raised. We believe these changes enhance the overall quality and comprehensiveness of the appendix.
> >
> > **Figure 5 and comparison with ChemCPA.**
> >
> > Thank you for the comment. We conducted additional analyses comparing the results in Figure 5 to another baseline (ChemCPA). We added this experiment to our appendix A.6.1.
> >
> > As shown, our proposed method is more robust across drug clusters compared to the baseline, where some clusters report significantly lower performance.
> >
> > We believe this result further strengthens the robustness of the proposed approach.

---

> > > ### Author Response · Authors · 2024-11-20
> > > **Rebuttal to Reviewers' Comments (part 3)**
> > >
> > > **Does the adapter fit well with other types of foundation models aside from scGPT?**
> > >
> > > Thank you for your question. Our proposed model is designed to be applicable to all transformer-based foundation models, not just scGPT (we mention this aspect at the beginning of Section 3). Indeed, the only architectural requirement of the proposed approach is a transformer-based architecture, where our conditional adapter is inserted at each transformer block (see Figure 1).
> > >
> > > We showcased our approach based on scGPT, as it is a published method with open-source code that follows a traditional transformer architecture. Other models, such as Yang et al, 2022 (scBERT), Theodoris et al., 2023 (Geneformer) and Hao et al., 2024 (scFoundation), follow a very similar approach, with minor modifications in the featurization and training dynamics.
> > >
> > > We acknowledge the importance of demonstrating the results on other architectures, which will be the subject of future research. However, the main contribution of this work is to demonstrate the ability of a fine-tuned single-cell FM to perform molecular perturbation prediction, which we showed across multiple settings, analysis, and against state-of-the-art baselines.
> > >
> > > **“In Sec 3.1, lines 209-214 I struggle to understand the logic…”**
> > >
> > > We apologize for the confusion. The second half of the paragraph was meant to indicate that for training and evaluation purposes, single-cell profiles corresponding to cell-line-drug-pairs are aggregated to average (or "pseudobulk") measurements. As you have suggested, we added Section A.2 to the Appendix, which gives a full overview of the model to help clarify inputs, outputs, and the model architecture.
> > >
> > > **it is not very clear what objective is used in 3.2.**
> > >
> > > We apologize for any confusion. In Section 3.2, we formulate the problem as a regression task. Specifically, we input the (average) control expression, the molecule embedding, and the gene token into our model. To evaluate the correctness of the prediction, we calculate the mean squared error (MSE) between the predicted perturbation and the true perturbations. Equation 3 outlines the loss function used, and in this context, it ensures that the model accurately perturbs the average control expression.
> > >
> > > We will clarify this aspect further in the paper to enhance understanding. We also added Section A.2 to the Appendix, which further clarifies the overall training procedure and architecture.
> > >
> > > **Is the evaluation carried out to the 20 top DEG or also the prediction?**
> > >
> > > Similar to previous studies (e.g., ChemCPA), we train our model on the entire set of genes. For the evaluation, we report results on the top 20 DEGs. This focus on the top 20 DEGs is intentional, as it represents a more challenging and interesting task, providing a clearer demonstration of our model's capability to identify the most significant changes. Additionally, we provide Figure 8 in the Appendix, which shows the performance of our model for different DEG sizes.
> > >
> > > **Are all baselines able to perform all tasks or do you have to adapt them?**
> > >
> > > Thank you for your question. Most of the baselines are able to perform all tasks with minimal adaptation on our part. However, chemCPA is not able to perform the unseen cell type zero-shot task (and therefore, results have not been included for this specific model-setting combination). This limitation arises because chemCPA's architecture requires cell type embeddings that are learned during training, which are not available for unseen cell types.

---

> > > > ### Comment · Reviewer_Le4o · 2024-11-24
> > > > **Answer to the rebuttals**
> > > >
> > > > Dear authors,
> > > >
> > > > First, I appreciate your work and time invested in answering my questions. Your answers were thorough and some of my doubts have been cleared. I am really sorry I cannot be more positive in my assessment; I reread the paper and I still find some aspects problematic beyond the extent of the rebuttal. I will explain myself in what follows.
> > > >
> > > > > The choice of the mean embeddings.
> > > >
> > > > I still believe that the prediction of perturbation effects on the mean embeddings of cell lines is a very strong limitation of this work. At the beginning I thought $c$ indicated "cell type", but now that I better examine sciplex3 I realize the model has been tested at a cell-line level. In my opinion, this resolution is way too coarse for this setting to be useful, there is so much underlying biology within a cell line that goes unconsidered here. In future iterations of the work, I would recommend using a dataset with finer annotation.
> > > >
> > > > In a way, predicting the average effect defies the purpose of doing single-cell in the first place, when you could predict everything on bulk datasets with way more compounds. Here you are constraining yourself to sciplex3 that only has 188 drugs, which is for me insufficient proof of generalization to the chemical space.
> > > >
> > > > > The use of sciplex3
> > > >
> > > > I would like to go more in-depth into this. In a dataset with 188 drugs and 3 cell lines, it is for me really unlikely that a model is capable of performing ood predictions, especially when leaving out a wealth of 30% of drugs for testing. I know sciplex3 well myself and the signal is not very strong per compound. More in detail, most of the compounds exhibit roughly no shift from controls, suggesting that evaluating the $R^2$ for the predictions to these compounds is not very informative, as probably already controls have a high $R^2$ with their mean response vector (did you check this aspect?). I think this aspect is not only problematic with this paper but in the field in general, though I have to highlight it here.
> > > >
> > > > > Baselines
> > > >
> > > > All the baselines you consider act at a single-cell level. How do you make the comparison fair? Do you predict with them and then average their prediction to compare it with the actual mean of the perturbed cells? I think this is not a very fair comparison, since the task shift and claims of such models are completely different as they act on a way finer level.
> > > >
> > > > > Comparison with chemCPA
> > > >
> > > > I browsed the repo of chemCPA and did find the ood drugs annotated in the `.obs` field of their AnnData, in case you want to check for future revisions of your work.
> > > >
> > > > As said, I feel I cannot go for a score raise here, but I do believe this work has a future potential if expanded and further researched in more aspects.
> > > >
> > > > Best regards,
> > > >
> > > > The reviewer

---

> > > > > ### Author Response · Authors · 2024-11-26
> > > > >
> > > > > Thank you for recognizing the time and effort we dedicated to addressing your previous comments. Regarding the new points you have raised, please find our responses below.
> > > > >
> > > > > **I still believe that the prediction of perturbation effects on the mean embeddings of cell lines is a very strong limitation of this work. ... In my opinion, this resolution is way too coarse for this setting to be useful, there is so much underlying biology within a cell line that goes unconsidered here. In future iterations of the work, I would recommend using a dataset with finer annotation.**
> > > > >
> > > > > There is no ground truth data available where compound responses are paired at the single-cell level. Consequently, comparisons must be made using quantities derived from distributions. Among these, the mean is the simplest and most intuitive metric. This approach has also been adopted by previous studies addressing similar challenges, *ChemCPA* [1] and *bioLORD* [2].
> > > > >
> > > > > We respectfully disagree with the reviewer's expectation of significant heterogeneity within a cell line, especially when only considering negative control perturbations. While there is usually some heterogeneity observed at the single-cell level, the strongest drivers are ubiquitous biological processes such as variations in a cell's cycling state. The phenomenon has been investigated in the literature, such as Fig. 2h of [3]. Therefore, we would not expect each of the tool cell lines considered in sciplex3 to exhibit a heterogeneity approaching that of different cell types in human primary cells.
> > > > >
> > > > > As far as encoding the control cell distribution is concerned, we would like to reiterate that our simple approach outperforms the current state-of-the-art. We also tried running our method with single-cell input data, pairing perturbed cells randomly with control cells given the lack of paired data. *We observed performance comparable with our reported metrics, but significantly increased training times*, motivating us to stick with the simpler approach presented in the paper. Training on the averaged data is also in line with prior work, such as BioLord [2]. While other approaches to encoding the control distribution would certainly be an interesting area for future research, we consider it out-of-scope for this publication.
> > > > >
> > > > > To summarize, we evaluate our model in line with prior publications on the subject and in a way to accommodate inherent pairing limitations of the data. Doing so, we find that our encoding approach is sufficient to outperform state-of-the-art methods.
> > > > >
> > > > >
> > > > > **The use of sciplex3, most of the compounds exhibit roughly no shift from controls, (did you check this aspect?). I think this aspect is not only problematic with this paper but in the field in general.**
> > > > >
> > > > >
> > > > > While it is true that this dataset is relatively small, It remains the largest publicly available single-cell dataset of chemical perturbations, and it is the sole dataset of this kind used by related prior work, such as ChemCPA [1] and bioLORD [2]. Even if we expect to have access to larger and more comprehensive datasets in the future, sample multiplexing techniques are both expensive and time-consuming, and therefore we expect few-shot and OOD questions to impact future datasets, as well.
> > > > >
> > > > > Moreover, the fact that the compound signal is not very strong *is actually an important challenge characterizing this application*, with methods such as ours tackling it. Nonetheless, it has been shown that the signal included in this dataset is statistically significant, not only from the original publication but also from follow-up studies, for example, scPerturb (Nature Biotechnology, 2024) [4], which comments:
> > > > > 	“The most extensive drug dataset is sci-Plex 3, which includes 188 drugs tested across three cell lines; 107 of those perturbations had significant effects on cell states according to E-test analysis (Supplementary Table 3).”
> > > > >
> > > > >
> > > > > [1] Hetzel et al. Predicting cellular responses to novel drug perturbations at a single-cell resolution. NeurIPS. 2022
> > > > >
> > > > > [2] Piran et al. Disentanglement of single-cell data with biolord. Nature Biotechnology. 2024.
> > > > >
> > > > > [3] Ursu et al. Massively parallel phenotyping of coding variants in cancer with Perturb-seq. Nature Biotechnology. 2022.
> > > > >
> > > > > [4] Peidli et al. scPerturb: harmonized single-cell perturbation data. Nature Biotechnology. 2024.

---

> > > > > > ### Author Response · Authors · 2024-11-26
> > > > > >
> > > > > > **All the baselines you consider act at a single-cell level. How do you make the comparison fair?**
> > > > > >
> > > > > >  We believe that our comparison between methods is fair because our evaluation pipeline is the same as that of the prior work focused on chemical perturbations we compared against (ChemCPA [1] and BioLORD [2]). Additionally, we strove to use the considered methods as originally intended. In particular:
> > > > > > We trained ChemCPA at the single-cell level for training, observing a performance drop when we tried changing it to train on average perturbation responses.
> > > > > > We trained BioLORD at the average perturbation response level, which is in line with their published method.
> > > > > > SAMS-VAE was not initially designed for the prediction of chemical perturbation responses. We adapted it to the task by providing it with molecular embeddings.
> > > > > >
> > > > > > **Comparison with chemCPA**
> > > > > >
> > > > > > We thank the reviewer for going through the repo and pointing us to the splits. We would like to reiterate that our splits are very much comparable in spirit to the ones used in ChemCPA, just not identical. To investigate this more thoroughly, we evaluated scDCA on the `ood_drugs` split provided in the ChemCPA repo. In line with our expectations, we observe a performance comparable to the one we report on our "unseen drug" split (scDCA on ChemCPA split: R^2 = 0.86, scDCA on our "unseen drug" split: mean R^2 = 0.81), significantly better than the reported performance for ChemCPA (ChemCPA on ChemCPA split: R^2 = 0.64 reported in [1], ChemCPA on our "unseen drug" split: mean R^2 = 0.68).
> > > > > >
> > > > > > [1] Hetzel et al. Predicting cellular responses to novel drug perturbations at a single-cell resolution. NeurIPS. 2022
> > > > > >
> > > > > > [2] Piran et al. Disentanglement of single-cell data with biolord. Nature Biotechnology. 2024.
> > > > > >
> > > > > > [3] Ursu et al. Massively parallel phenotyping of coding variants in cancer with Perturb-seq. Nature Biotechnology. 2022.
> > > > > >
> > > > > > [4] Peidli et al. scPerturb: harmonized single-cell perturbation data. Nature Biotechnology. 2024.

---

> > > > > > > ### Comment · Reviewer_Le4o · 2024-11-26
> > > > > > >
> > > > > > > ## Dear Authors,
> > > > > > >
> > > > > > > First and foremost, I commend you for the effort you have put into defending your work. I deeply appreciate the extensiveness of your response and the clarity with which you have presented your line of thought. However, I must reiterate my position and maintain that I believe the paper, in its current form, is not ready for publication at ICLR. Once again, I regret having to provide such a critical assessment.
> > > > > > >
> > > > > > > ### The Significance of the Mean Embedding
> > > > > > >
> > > > > > > I must respectfully disagree with the authors’ downplaying of the heterogeneity within control cells across cell lines in the sciplex3 dataset. Plotting the 2D embedding of control cells alone captures a significant portion of intra-cell line variability, whereas the perturbation effect tends to be weak and sparse. The argument that intra-cell line heterogeneity is lower than in primary cells does not, in my view, adequately justify disregarding this variability. Moreover, referencing a publication on a dataset different from sciplex3 to substantiate your claim seems misplaced. I tried to plot the cell cycle scores in sciplex3 and, while there is indeed some variability explained by the cycling states of a cell, that does not seem to have such a clear pattern as the publication you reference.
> > > > > > >
> > > > > > > In light of emerging methods that operate at the distribution level, such as optimal transport, I believe that mean-based prediction is limited—particularly with the advent of more complex datasets.
> > > > > > >
> > > > > > > ### Predicting the Mean of a Perturbation Vector
> > > > > > >
> > > > > > > As I understand it, the task defined in the paper fundamentally involves a regression from the control mean of each cell line to the perturbed mean of each cell line, conditioned on a perturbation embedding. This raises the question: why is scGPT necessary for this task? Wouldn’t a simpler model suffice? (To clarify, I am not suggesting you perform additional experiments given time constraints.) My concern is that the benefits of employing a foundation model for this specific task remain underexplored.
> > > > > > >
> > > > > > > ### The Evaluation Approach
> > > > > > >
> > > > > > > Building on my earlier critique, I reiterate my concern about the evaluation methodology. Using $R^2$ as a metric in this context can be misleading. In cases where perturbations result in negligible deviations from controls, the means are likely to be close. As such, predicting the controls could yield a high $R^2$, which might not genuinely reflect the model's efficacy. This issue represents my primary concern with the paper and, to some extent, with the field as a whole.
> > > > > > >
> > > > > > > ### The Limits of sciplex3
> > > > > > >
> > > > > > > I also find the justification for the evaluation pipeline, given the limitations of sciplex3, unconvincing. The claim that sciplex3’s small dataset size can enable generalization across chemical space seems tenuous—especially considering that the relationship between chemical structure and phenotypic response is not one-to-one. Additionally, in the referenced paper, the E-test denotes energy test, which measures distances between distributions rather than means. If my understanding is correct, this strengthens my argument against a mean-based approach.
> > > > > > >
> > > > > > > ### Comparison with chemCPA
> > > > > > >
> > > > > > > It is reasonable to expect that training a VAE model on means alone would yield suboptimal results, which may explain the poor performance of chemCPA in your comparison. However, I remain unconvinced by the comparison between sams-VAE, chemCPA, and scDCA. These methods do not explicitly address the regression task of predicting one variable from another conditionally, making the comparison less compelling.
> > > > > > >
> > > > > > > ### Final Remarks on the Overcoming of Baselines
> > > > > > >
> > > > > > > Unfortunately, while I acknowledge the improved performance over the baselines in terms of $R^2$ prediction (again, a flawed metric to me when predicting the means), I do not think that this is enough to convince me that the task is biologically relevant. Although this is an ML paper with an interesting intuition, all reasons listed so far limit its biological impact to me.
> > > > > > >
> > > > > > > ### Conclusion
> > > > > > >
> > > > > > > As a result, I have decided to maintain my score. I sincerely hope this feedback is not perceived as an unmotivated barrier, as I genuinely appreciate the authors’ efforts in presenting their perspectives.
> > > > > > >
> > > > > > > Best regards,
> > > > > > >
> > > > > > > The Reviewer

---

> > > > > > > > ### Author Response · Authors · 2024-12-04
> > > > > > > >
> > > > > > > > From your latest reply, we understand that our analysis remains unconvincing regarding the points raised, particularly the choice of datasets and metrics used to evaluate the work. As we have explained in our previous responses, we adhered to the same metrics, datasets, and experimental settings utilized in prior works in the field. This represents the extent of analysis that can be conducted given the current unavailability of additional data.
> > > > > > > >
> > > > > > > > We remain hopeful that the release of additional datasets in the future will help address these concerns more effectively.

---

### Official Review · Reviewer_Qies · 2024-11-03

**Soundness:** 3
**Presentation:** 3
**Contribution:** 3
**Rating:** 6
**Confidence:** 3

**Summary:**

This work considers the molecular perturbation prediction by leveraging Foundational Models (FMs). More specifically, it introduces scDCA, which utilizes the pretrained FM, scGPT, that is trained on single cell omic data. In this design, to enforce the molecular information, an adapter layer has been introduced to the model which allows for an efficient fine tuning of the model by keeping the FM’s weights frozen, while allowing for considering an extra modality.

**Strengths:**

1.	Enables fine-tuning with a limited paired data by training only the drug conditional adapter layer.
2.	By using the adapter and ChemBERTa, a molecular FM, scDCA enables the utilization of different modalities information due to its design.
3.	It seems that based on the provided results, the proposed framework leads to superior performance with respect to baselines.

**Weaknesses:**

1.	One of the challenges that this study considers is data scarcity. Although there are many recently proposed parameter efficient fine-tuning approaches, they are not considered in evaluations (table 1).
2.	There is no clear explanation on canonical projection.
3.	The diagram in figure 1 does not match with the projections in equation 5 (based on the diagram, for both $h^{down}$  and $h^{up}$  use the projection of $b_l$ ; however, in equation for $h^{up}$ only $b_l$ is in the summation). In general, it is not obvious if projection is referring to module $f$ or the canonical projection.
4.	The quality of figure 7 is very bad.

**Questions:**

1.	Why other FMs are not considered in this work? Comparison with other single cell FMs can provide more insights in the benefits of the proposed adapter architecture.
2.	Why other parameter efficient fine-tuning methods are not considered in the evaluation? This could provide more insight into the capabilities of scDCA specifically in capturing the domain shift.
3.	Is it possible to do an ablation study, specifically to show the effect of Resnet more clearly?
4.	Is it possible to explain why in figure 4 for some genes the predicted expressions are out of the true expression regions?
5.	Again, in figure 4, for many genes the predicted expressions are close to the ends of the True expression regions. Is it possible to provide an evaluation of framework’s reliability and limitations?

---

> ### Author Response · Authors · 2024-11-22
> **Rebuttal to Reviewers' Comments**
>
> Thank you for your insightful comments and for your recognition of our work’s significance and performance.
>
> **Comparison with other efficient fine-tuning approaches**
>
> Thank you for your feedback. It is true that efficient fine-tuning approaches that address data scarcity have recently been proposed. However, our study tackles a specific multi-modal challenge: fine-tuning a model pre-trained on data from X, where X is gene expression data in our application, through paired data (X, Y, X’), where Y are molecular structures, and (X, X’) correspond to gene expression before/after the perturbation in our application. The proposed approach can effectively handle this multi-modal finetuning setting, ensuring that both gene expression data and molecular information are integrated seamlessly and addressing both data scarcity and multi-modality simultaneously.
> Previous works have not addressed this specific setting. Therefore, comparisons with other fine-tuning strategies would require non-trivial method development.
>
> Additionally, in this study, we are not primarily focused on proposing a new fine-tuning strategy. Rather, we are interested in extending and customizing a consolidated approach (adapter-based finetuning) to enable an important application (molecular perturbation prediction).  Given our main goal, we focused on the comparisons with other molecular perturbation prediction methods recently proposed in the literature.
>
> We appreciate your suggestion and will consider evaluating additional parameter-efficient fine-tuning approaches in future work to further enhance our methodology.
>
> **There is no clear explanation on canonical projection.**
>
> Thank you for your input. The goal of the projection applied to the molecular embedding is to transform it into the same dimension $d_{bottleneck}$ used for the bottleneck layer. We realize the term “canonical projection” could be confusing if not formally defined, so we removed it in the revised version of the manuscript, simply using “projection”.
>
> **The diagram in figure 1 does not match with the projections in equation 5**
>
> To better clarify and summarize the architecture, we have added details to Section A.2 in the Appendix, including all the steps and components of the scDCA architecture. While Figure 1 is accurate, we omit some details from it to improve clarity. In Figure 1, the projection after the molecule embedding transforms it into the same dimension  $d_{bottleneck}$ used for the bottleneck layer.
>
> **The quality of figure 7**
>
> Thank you for pointing this out. We have fixed the figure quality in the revised version.
>
> Why other FMs are not considered in this work?
> We acknowledge the importance of demonstrating the results on other architectures, which will be the subject of future research. However, the main contribution of this work is to demonstrate the ability of a fine-tuned single-cell FM to perform molecular perturbation prediction, which we showed across multiple settings, analysis, and against state-of-the-art baselines.
> Importantly,  our proposed model is designed to be applicable to all transformer-based foundation models, not just scGPT. Indeed, the only architectural requirement of the proposed approach is a transformer-based architecture, where our conditional adapter is inserted at each transformer block.
>
> **Is it possible to do an ablation study, specifically to show the effect of Resnet more clearly?**
>
> Following the suggestion, we have conducted this experiment and included it in Table 2 in Section A.3 in the Appendix. As we have illustrated, res_net improves the performance of our model in all settings.
>
> **Is it possible to explain why in figure 4 for some genes the predicted expressions are out of the true expression regions?**
>
> Thank you for your question. Given that these are model predictions, it is possible that they are less accurate for specific genes associated with specific perturbations. Additionally, the inherent biological variability in gene expression data can contribute to these differences.
>
> Our primary goal with this figure is to show that, in many cases, the model’s predictions successfully capture log-fold changes up to the standard deviation of the underlying single-cell distribution. Even if this is not always true, we showed that, overall, the proposed model outperforms all the considered baseline.

---

> > ### Author Response · Authors · 2024-11-22
> > **Rebuttal to Reviewers' Comments (part 2)**
> >
> > **In figure 4, for many genes the predicted expressions are close to the ends**
> >
> > Thank you for raising this point. We have included a discussion of the limitations of our work in Section 5 of the manuscript. Additionally, to more robustly assess the performance of our method, we have added t-test results in Section A.8 of the Appendix, which demonstrate statistically significant improvements over all baselines, further strengthening its reliability.
> > We acknowledge that for some genes, the predicted expressions are close to the ends of the true expression regions. This may be due to several factors, including inherent biological variability, data scarcity, and the complexity of certain gene expression patterns. We will continue to refine our model in future work to improve its accuracy and robustness, particularly for challenging gene expression patterns.

---

> ### Comment · Reviewer_Qies · 2024-11-25
> **Response to Rebuttal**
>
> Thank you for your efforts and the time you dedicated to addressing my concerns and questions. While some of my concerns have been resolved, I regret that I am unable to increase my score for the submission. Regarding your rebuttal, particularly as part of the focus is on efficient fine-tuning, I continue to feel that the study lacks a comparison with other efficient fine-tuning methods. I tried to elaborate on my perspective below.
>
> I am unclear why a comparison with other efficient fine-tuning approaches is considered non-trivial, especially given that you have already fine-tuned scGPT. If fine-tuning scGPT is feasible, implementing other efficient fine-tuning algorithms should not present significant challenges. Moreover, without a comparison with other efficient fine-tuning methods, the significance of the adapter module is not sufficiently emphasized. I hope you will consider addressing this point in your future work.
>
> Thank you once again.

---

> > ### Author Response · Authors · 2024-12-04
> >
> > We sincerely thank you for the time and effort invested in reviewing our paper, as well as for the constructive and valuable feedback. We appreciate your supportive comments and the fact that the rebuttal helped resolve some of them.
> >
> > Regarding the comparison with other efficient fine-tuning approaches, the main complexity is in the fact that our application differs significantly from standard finetuning settings. For instance, our adapter layers are conditioned on a modality different from the one used during pre-training and fine-tuning. This makes direct comparisons with other “off-the-shelf” models non-trivial, as they would require adaptation to our application or assumptions about some aspects of the model.
> >
> > We fully agree that extending this work to include other fine-tuning strategies tailored to our application is a promising avenue for future research, and we intend to explore this direction. Still, we believe our contribution, being the first work successfully leveraging fine-tuning of single-cell FMs for this task, represents an important step for the field.

---

### Official Review · Reviewer_hmPT · 2024-11-04

**Soundness:** 3
**Presentation:** 3
**Contribution:** 3
**Rating:** 8
**Confidence:** 4

**Summary:**

The authors introduce single-cell drug-conditional adaptor (scDCA), an approach to fine-tune existing single-cell foundation models to predict the effect of drug perturbations in various cell-types in a parameter efficient way. scDCA accomplishes this by freezing the foundation model, scGPT in this case, and incorporating a drug-conditional adaptor in each layer. In doing so the authors show scDCA can generalize to unknown drug perturbation and cell types.

**Strengths:**

Work is original, the idea of injecting molecular embeddings in a way that stops catastrophic forgetting in single-cell foundation models is interesting and has not, to my knowledge, been shown before. Work is easy to follow and well-explained and addresses an important problem in predicting gene expression. If we had a model that could predict the effect in gene expression of drugs, it would revolutionize drug discovery. This model is the first step towards this reality.

**Weaknesses:**

I have a list of concerns that I would like to be addressed before I can accept this paper. They are listed below:

1. Figure 2 needs to be improved. I understand that the icons to the left are different cell types but they should be labeled, I see the neuron in the middle but what are the other two? Also what does each square represent a data point? I understand what you are trying to show but it can be visualized better, the figure is confusing as it stands now.
2. Why not take the output of the scGPT plus the output of the molecular embedding, concatenate it to predict the effect of expression. Is this the experiment shown in section A.2?
3. "By featurizing cell types based on their initial gene expression, our framework can generalize to unseen cell types in both zero-shot and few-shot settings." Line 328. This is not really a result of the framework, it is the result of using the scGPT and doing this efficient fine-tuning. The generalization to different cell-types arises from keeping the knowledge from the pretrained foundation model, the framework does not add anything to the pretrained model from the perspective of cell-types.
4. There needs to be statistical tests for Figure 3, there is large overlap in the error-bars. A t-test to show the difference is significant is necessary to deem if there is actual improvement.
5. It does not make sense that in Table 1 few-shot cell-type prediction is an easier task than unseen drug covariate.
6. What is Figure 7? What does DEG stand for? What is it showing?
7. Arguably the most important experiment of this paper is in scDCA predictions are robust across different targets, but the authors are vague in the description of the experiment and their results. Specifically what is meant by this phrase: "We grouped R2 for each target-cluster"?
8. I believe the claim that the model generalizes to novel cell-types is a bit exaggerated. The cell-types assessed in this paper are two breast cancer cell-lines and one leukemia cell-line. Did the authors make sure that the breast cancer cell-lines did not appear in both the train and test set in the cell-type splits because even though they have different names they would be very similar in expression profile.
9. For the unseen drugs how similar are the drugs to each other? Can there be a plot of Tanimoto similarity of the drug-similarity in the splits. If the drugs are structurally very similar then the model can overfit on that.
10. One of the biggest problems in this paper is the overlap between scGPT pertaining set and this test set. Has scGPT already seen the gene expression vectors of these perturbation experiments? Could this not make the results seem much better than they are? This could also explain why the fine-tuning did so poorly, because scGPT had forgotten information it had already seen. Even if scGPT did not see exactly the same vectors it could have seen vectors that are very similar.

**Questions:**

1. I am confused by the terminology "covariate"? What is a covariate? Cell-type? I have not seen this word used in single-cell literature and ask the authors to clarify its meaning. If it means cell-type why not say cell-type?

---

> ### Author Response · Authors · 2024-11-20
> **Rebuttal to Reviewers' Comments (part 1)**
>
> Thank you for your detailed comments. We truly appreciate that you found the proposed approach original, novel, well-explained, and, especially, a first step toward an impactful application.
> We have addressed all the points raised below, including providing additional analysis, figures, and edits to the manuscript.
>
> **Improving Figure 2**
>
> Thank you for pointing out this matter. We have taken your suggestions into account and improved Figure 2 accordingly.
> Cell types in this figure are for illustrative purposes, and each square represents a data point (for reference, the cell types used in this study are included in Table 8 of the Appendix). Our main goal for this figure is to introduce the 4 different generalization tasks.
> We changed cell type icons to avoid confusion, and we added labels to clearly identify the axes. Additionally, we enhanced the caption.
>
> **Baseline using the concatenated output of scGPT and the molecular embedding**
>
> Thank you for your comments and suggestions. Following the suggestion, we have conducted an experiment where the output of scGPT is concatenated with the output of the molecular embedding to predict the effect on gene expression. The results of this experiment have been added to Table 2 in Section A.3 in the Appendix. Our findings indicate that scDCA outperformed this baseline on all tasks. Additionally, it is worth noting that our approach is also superior in terms of the number of parameters, using less than 1% of the parameters compared to the proposed model.
> These results are different from the experiment shown in Section A.4 (previously A.2). Specifically, we included this table because, as mentioned in Section 3.2, one common approach to fine-tuning foundation models (FMs) is feature-based fine-tuning. Our results demonstrate that this approach does not perform well in our context, highlighting the superiority of our proposed method.
>
> **"By featurizing cell types based on their initial gene expression, our framework can generalize to unseen cell types…”**
>
> Thank you for your comment. We agree that this capability really emerges from leveraging a single-cell foundation model (scGPT), which has been pre-trained across multiple cell types and conditions.
> At the same time, the goal of the proposed approach is exactly that of efficiently fine-tuning the FM such that the information about cell types is preserved and effectively leveraged for molecular perturbation prediction.
> We agree that the introduced framework does not add anything to the pre-trained model from the perspective of cell types. However, being able to preserve such information, while allowing the incorporation of molecular representations and the ability to flexibly integrate them into the pre-trained knowledge is not trivial, and it is exactly the problem that the proposed framework addresses. As we have shown, other finetuning strategies (Appendix A.3 and A.4) do not lead to the same results as the proposed approach that we attribute to the introduced elements.
>
> We further clarified this point by adding a sentence in the main text after the one you highlighted:
> “Importantly, the ability to generalize to new cell lines arises from leveraging knowledge available in the pre-trained model and scDCA preserving and integrating it with molecular representations during finetuning.”
>
> **Statistical tests for Figure 3**
>
> Thank you for your helpful suggestion. We have added the results of statistical tests in Appendix A.8 (Tables 4, 5, 6, 7). Overall, we found that all the comparisons are significant at a 95% nominal confidence level.

---

> ### Author Response · Authors · 2024-11-20
> **Rebuttal to Reviewers' Comments (part 2)**
>
> **Difficulty of different tasks in Table 1**
>
> Thank you for your observation. While the different performance levels between few-shot cell-type prediction and unseen drug-covariate [renamed unseen drug-cell-line in the revised manuscript] prediction may seem surprising, these are fundamentally different tasks, each with its own unique challenges and complexities.
> In particular, in both tasks, many drug-cell-line pairs are observed. However, for the drug-cell-line combination task, we split the data 50% train/50% test (280 train, 281 test combos) while for the few-shot task, the drug measurements in the held out cell line are split 10% train/90% test (392 train, 169 test combos). This creates specific challenges for the model in the few-shot cell-type generalization, with a much smaller percentage of drugs observed in the held out cell line (~50% vs 10%). There might also be additional sources of variability that make the direct comparison challenging, such as the impact of the different top-20 DEGs for each drug-cell-line combination, which can lead to task-dependent difficulty.  Therefore, direct comparisons of the prediction results between these tasks are not appropriate.
> In this study, our main goals are to assess (1) the robustness of the proposed method across different tasks, and (2) the superiority of the proposed method against all the baselines across different tasks. We believe that our results clearly demonstrate both goals. While our results can also offer some insights into task differences, we believe that direct comparisons are not conclusive. We clarified this aspect by adding a sentence in Section 4.2:
> “While we observe scDCA results to be robust across the different tasks, direct comparison of the performance across tasks is challenging, given differences in splitting strategies and the unique features of each task.”
> We also added a clarification about the different splitting ratios to the Appendix, Section A.1.
>
> **What is Figure 7? What does DEG stand for? What is it showing?**
>
> Thank you for your question. DEGs stands for differentially expressed genes. In the presence of a new drug, not all genes may exhibit changes in expression levels, and a subset of genes may exhibit significant changes. Reporting the results on all genes may not capture the drug-related signal effectively. Therefore, similarly to previous studies [1,2], we focus on the DEGs, as these are the genes that show the most significant changes and are therefore more interesting in assessing the model’s performance. While we focus on a specific number (20) of DEGs in the main results, for completeness, Figure 8 (previously Figure 7)  illustrates the performance of our model in predicting DEGs for different sizes. The size of the DEG set indicates the difficulty of the task, with smaller sets representing more challenging prediction scenarios. We have clarified this in the figure caption and the corresponding text in the paper.
> [1] L. Hetzel et al. Predicting cellular responses to novel drug perturbations at a single-cell resolution, NeurIPS, 2022.
> [2] Y. Roohani et al. Predicting transcriptional outcomes of novel multigene perturbations with gears. Nature Biotechnology, 2024.
>
> **Clarifying target-cluster results (Figure 5)**
>
> Thank you for your question. The goal of this experiment is to understand the heterogeneity of prediction quality. To achieve this, we stratify the results according to clusters defined by target annotations.
> More specifically, the design of the experiment is as follows:
> We first obtain compound annotations from ChEMBL.
> We conduct a compound hold-out (unseen drug) experiment, ensuring that all annotated compounds are reserved for the test set.
> For evaluation, we subset the test set compounds to one target cluster at a time.
> We then report the R2 values for each target cluster.
> By grouping the R2 values for each target cluster, we aim to assess the robustness of scDCA predictions across different targets. This approach allows us to evaluate the prediction quality heterogeneity for compounds targeting different biological pathways or mechanisms.
> We clarified this in Section A.6.1 in the Appendix.

---

> ### Author Response · Authors · 2024-11-20
> **Rebuttal to Reviewers' Comments (part 3)**
>
> **Generalization to new cell types and similarity between cell types**
>
> In fact, the sciplex3 dataset is composed of the cell lines A549 (lung adenocarcinoma), K562 (myelogenous leukemia) and MCF7 (breast adenocarcinoma). Since there are only these three cell lines present in the sciplex3 dataset, for cell-type generalization tasks, we trained the model in a leave-one-out manner (which corresponds to three splits in total)
> We added the results for scDCA and baselines for each split separately in Table 8 of Appendix A.9. The lowest performing cell line holdout for scDCA is in fact K562, but this trend is not evident for the baselines (BioLord and SAMS_VAE). In every split, scDCA still outperforms the baselines.
> In general, we would expect the difficulty of the unseen cell line task to vary according to the degree of biological similarity of the cell lines, but the sciplex3 dataset only offers limited data to further investigate this question. We have added this as a limitation to the conclusion.
>
> **“For the unseen drugs how similar are the drugs to each other?”**
>
> Thank you for your insightful comment. Following the suggestion, we analyzed the similarity between training and test molecules, focusing on the generalization to new molecules.
> For each test molecule, we computed its maximum similarity to any training molecule. We used standard functions, leveraging Tanimoto similarity on Morgan fingerprints (radius=2, num_bits=2048) as implemented in RDKit library.
> We show the distribution of maximum similarities in Figure 9 in A.10.
> As shown, the vast majority of molecules have low similarity to training structures, with 88% of them having a similarity ≤ 0.4 to any training molecule (0.4 is often considered the threshold to define structural novelty [1]).
>
> [1] Dalke, The chemfp project. J Cheminform 11, 76 (2019).
>
> **Overlap between scGPT pretraining set and test set**
>
> Thank you for raising this concern. Given that scGPT does not release the full list of datasets used for pretraining, we have thoroughly examined the scGPT GitHub repository, focusing on the functions used to build the dataset that scGPT was trained on. To the best of our knowledge, scGPT was trained on a set of tissues with no cell lines or perturbed gene expression. Therefore, we believe that there is no overlap between the dataset used for training scGPT and the perturbed dataset we used for evaluating our method.
> Additionally, we noticed that scGPT itself in the original paper fine-tunes the model to predict genetic perturbations using cell line datasets.
>
> **I am confused by the terminology "covariate"?**
>
> We thank the reviewer for bringing this to our attention. We initially used this term because it was used in prior work, such as chemCPA. The term "covariate" is used broadly for any kind of additional information about a single cell such as different cell types, cell states, diseases, etc. Indeed,  in our paper, we mainly used it to refer to a cell type. However, we understand that this may cause confusion, especially since it is not handled as a categorical covariate in scDCA, but instead entirely encoded in the gene expression values of the control cells. For this reason, we changed the terminology in the paper to cell line/cell type wherever this is appropriate. This is currently implemented in the text; for time reasons, we will update the figures later as part of the overall revision/rebuttal process.

---

> > ### Comment · Reviewer_hmPT · 2024-11-25
> > **Response to Author Rebuttal**
> >
> > I thank the authors for their detailed response to my questions. All my concerns have been addressed; I have updated my score to an accept.

---

> > > ### Author Response · Authors · 2024-11-26
> > >
> > > Thank you for your thoughtful comments and for taking the time to review our work. We sincerely appreciate your feedback, which helped us improve the manuscript. We are glad that our responses addressed your concerns, and we are grateful for your updated recommendation.

---

### Official Review · Reviewer_TaaQ · 2024-11-10

**Soundness:** 3
**Presentation:** 3
**Contribution:** 2
**Rating:** 6
**Confidence:** 4

**Summary:**

The authors explore the challenge of predicting cellular transcriptional responses to novel molecular perturbations, a critical task in drug discovery. By leveraging pre-trained single-cell foundation models on large-scale datasets, the authors propose an efficient fine-tuning method using a drug-conditional adapter (scDCA). This approach utilizes less than 1% of the original model's parameters, allowing for efficient adaptation to novel drugs and cell types, even in zero-shot settings. The paper demonstrates that scDCA significantly outperforms existing baselines in various prediction tasks, including unseen drug-covariate combinations and zero-shot cell type generalization.

**Strengths:**

The introduction of drug-conditional adapters is an innovative and efficient method for fine-tuning large single-cell FMs, which is particularly impressive given the reduction in trainable parameters. This efficient approach maintains the original biological representations, allowing the model to generalize to new cell types and drugs.


The authors effectively build upon the latest advancements in transfer learning and efficient model adaptation, such as prefix tuning and hypernetworks. The use of a transformer-based single-cell FM architecture (scGPT) combined with pre-trained molecular embeddings (ChemBERTa) demonstrates a sophisticated integration of modalities.

The paper is well-structured, with a thorough explanation of the scDCA architecture and the experimental setup. The inclusion of figures and tables to illustrate model performance across various tasks aids in understanding the results.

**Weaknesses:**

The experiments are primarily conducted on the sciplex3 dataset, which, although extensive, focuses on a limited number of cell lines and drugs. The generalizability of scDCA to broader datasets with more diverse cell types and perturbations remains to be demonstrated. If no single-cell dataset is available, how about using the LINCS L1000 dataset? The NIH LINCS L1000 dataset provides rich gene expression profiles for thousands of chemical perturbations across diverse cell lines. Its comprehensive scope and availability would significantly enhance the evaluation of the scDCA model, particularly for zero-shot and few-shot generalization tasks. Incorporating this dataset would validate the model’s robustness and broaden its applicability in predicting transcriptional responses to novel drugs (One caution when using this dataset is that it can be somewhat noisy). While the LINCS L1000 data offers a rich resource for bulk transcriptomic studies, *it may not directly align with the single-cell focus of the paper*. However, it could still be useful for comparison studies. It could be used to compare how predictions from bulk data differ from those based on single-cell data, highlighting the added value of single-cell resolution.


While the model achieves strong predictive performance, there is limited discussion on the biological insights derived from the model's predictions. Understanding which gene-gene interactions or molecular features drive the predictions could enhance the impact of the work for biological research.

**Questions:**

The paper focuses on in vitro transcriptional responses using single-cell RNA sequencing data. However, translating these predictions to in vivo or clinical settings, where cellular environments are significantly more complex, remains an open challenge. How feasible do you think it is to extend this model to such real-world scenarios?

---

> ### Author Response · Authors · 2024-11-22
> **Rebuttal to Reviewers' Comments**
>
> Thank you for your detailed and positive comments. We truly appreciate that you found the proposed approach innovative and the paper well-structured.
> We have addressed all the points raised below, including providing additional analysis, figures, and edits to the manuscript.
>
> **Extension to additional dataset (LINCS L1000)**
>
> Thank you for your comment. We believe that the evaluation of perturbation models, especially in the single-cell setting, is challenged by the limited size of publicly available data, which we noted in the revised conclusion of the manuscript.
>
> We would like to emphasize that related works such as ChemCPA and BioLORD have also only shown results for the sciplex3 dataset. We are committed to finding related datasets that could be applied to our model. However, as you mentioned, the LINCS L1000 dataset is known to be very noisy, and results could be harder to interpret.
>
> We are actively exploring other potential datasets, including LINCS, that could be used to evaluate our model more comprehensively. As more high-quality single-cell perturbation datasets become available, we plan to extend our evaluations to include these datasets to further validate and enhance our model.
>
> **Additional discussion on the biological insights**
>
> Thank you for your feedback. In Figure 5 in the paper, we cluster molecules based on their targets (extracted from ChEMBL) and demonstrate that scDCA is robust across all target clusters. Details of this analysis are included in Section A.6 of the Appendix. Even with its limitations, we believe that this analysis provides valuable insights by illustrating that our model can generalize across different biological pathways and mechanisms of action (MOA), something that was not investigated in previous works.
> In future work, we are going to focus further on exploring these biological insights, for example, understanding features driving the predictions and important gene-molecule interactions.
> Finally, we would like to emphasize that the primary goal of this paper is methodological, focusing on developing a method that could effectively fine-tune single-cell FMs for the task of molecular perturbation prediction. The biological applications and insights derived from this method are promising and will be a significant area of future research.
>
> **How feasible do you think it is to extend this model to such real-world scenarios?**
>
> Thank you for your comment. We acknowledge that translating in vitro transcriptional responses to in vivo or clinical settings is indeed a significant challenge due to the increased complexity of cellular environments in real-world scenarios. While our current model demonstrates strong performance with in vitro single-cell RNA sequencing data, extending it to in vivo or clinical settings will require adopting our model to the complex in vivo or clinical settings, access to diverse in vivo datasets, and collaborations with clinical researchers.
> Despite these challenges, we believe that our methodological advancements provide a solid foundation for future research aimed at translating these predictions to in vivo and clinical settings. In particular, we believe that bridging domain-specific foundation models and efficiently fine-tuning them, especially in multi-modal settings, will be critical to support these real-world scenarios, which are often characterized by extremely limited and highly multi-modal data. We are committed to exploring these directions and enhancing the model's applicability to real-world scenarios.

---

> > ### Comment · Reviewer_TaaQ · 2024-11-26
> >
> > Thank you for the detailed rebuttal addressing my concerns and providing clarifications.
> > I appreciate the additional analysis and insights presented, particularly regarding the clustering of molecules by their targets and the robustness across different mechanisms of action, as well as the emphasis on the methodological focus of the paper.
> > While your responses clarify several aspects, including the challenges of leveraging alternative datasets like LINCS L1000 and the potential for future exploration of biological insights and real-world applications, they do not fully mitigate the concerns regarding broader dataset evaluation and deeper biological interpretability.
> > Thus, I will maintain my current score as I believe the paper still demonstrates strong methodological innovation and potential but falls slightly short in addressing the broader applicability and biological depth comprehensively.

---

> > > ### Author Response · Authors · 2024-12-04
> > >
> > > We would like to sincerely thank you for your valuable and supportive feedback. We appreciate that our reply helped clarify the remaining concerns and that you found our paper delivering strong methodological innovation.
> > >
> > > We are committed to building on this foundation to explore broader biological questions in future studies. We fully agree that expanding the biological depth of this research is a critical and exciting goal for the future.

---

### Author Response · Authors · 2024-11-23
**Follow-Up on Responses and Revised Manuscript**

Dear reviewers, we would like to extend our sincere gratitude for your thorough review and valuable feedback on our manuscript.

We have carefully addressed all the concerns and questions raised in your reviews. This includes providing additional analysis, incorporating new figures, and making comprehensive edits to the manuscript. We believe these revisions have significantly improved the quality and clarity of our work.

We would greatly appreciate any further feedback you may have regarding our responses and the revised manuscript. Your insights are invaluable to us, and we are eager to address any additional questions or concerns you might have.

Thank you once again for your time and consideration.

---

### Meta-Review · Area_Chair_vN2b · 2024-12-19

**Metareview:**

The authors introduced single-cell drug-conditional adaptor (scDCA) to fine-tune the pre-trained single-cell foundation model scGPT for zero-shot perturbation prediction. The presented experimental results have shown scDCA can achieve better empirical performance than the existing models.

The authors have made significant effort to address the reviewers' questions and concerns during the discussion phases. However, no consensus was reached. Reviewer **Le4o** has serious concerns regarding the possible bias in empirical performance evaluation. Other borderline reviewers also have suggested the authors to compare with other fine-tuning methods, even if the architecture adaption can take time based on the authors' rebuttal, to better establish the significance of the proposed scDCA.

The reviewers have concerns, without comparing with other fine-tuning methods, on how significant is scDCA and what would be the main methodological contributions in the submitted work.

The authors may consider addressing these issues to improve the quality of their future submission. Especially, they may consider checking the following benchmarks to better establish the significance of their work:

1. https://github.com/aaronwtr/PertEval
2. https://arxiv.org/abs/2408.10609

**Additional Comments On Reviewer Discussion:**

During the discussion phases, the authors have significantly revised the original submission and provided additional experimental results to support their claims. However, no consensus was reached. One negative reviewer and some borderline reviewers still have concerns regarding the potential benchmark data and evaluation metrics. During the AC-reviewer discussion phase, the reviewers suggested that the authors should try to compare with existing fine-tuning methods and also other benchmarks beyond sciplex3 to avoid possible bias due to high similarity across samples.

---

### Decision · Program_Chairs · 2025-01-22

Reject